# Identifying rainfall-runoff events in discharge time series: A data-driven method based on information theory

Stephanie Thiesen[1], Paul Darscheid, Uwe Ehret[1]

[1]Institute of Water Resources and River Basin Management, Karlsruhe Institute of Technology - KIT, Karlsruhe, Germany

*Correspondence to*: Stephanie Thiesen (stephanie.thiesen@kit.edu)

**Abstract.** In this study, we propose a data-driven approach to automatically identify rainfall-runoff events in discharge time series. The core of the concept is to construct and apply discrete multivariate probability distributions to obtain probabilistic predictions of each time step being part of an event. The approach permits any data to serve as predictors, and it is non-parametric in the sense that it can handle any kind of relation between the predictor(s) and the target. Each choice of a particular

predictor data set is equivalent to formulating a model hypothesis. Among competing models, the best is found by comparing their predictive power in a training data set with user-classified events. For evaluation, we use measures from information theory such as Shannon Entropy and Conditional Entropy to select the best predictors and models and, additionally, measure the risk of overfitting via Cross Entropy and Kullback-Leibler Divergence. As all these measures are expressed in "bit", we can combine them to identify models with the best tradeoff between predictive power and robustness given the available data.

We applied the method to data from the Dornbirnerach catchment in Austria distinguishing three different model types: Models relying on discharge data, models using both discharge and precipitation data, and recursive models, i.e., models using their own predictions of a previous time step as an additional predictor. In the case study, the additional use of precipitation reduced predictive uncertainty only by a small amount, likely because the information provided by precipitation is already contained in the discharge data. More generally, we found that the robustness of a model quickly dropped with the increase in the number

of predictors used (an effect well known as the curse of dimensionality), such that in the end, the best model was a recursive one applying four predictors (three standard and one recursive): discharge from two distinct time steps, the relative magnitude of discharge compared to all discharge values in a surrounding 65-hour time window and event predictions from the previous time step. Applying the model reduced the uncertainty about event classification by 77.8 %, decreasing Conditional Entropy from 0.516 to 0.114 bits. To assess the quality of the proposed method, its results were binarized and validated through a

holdout method and then compared to a physically-based approach. The comparison showed a similar behavior of both models (both with accuracy near 90 %) and the cross-validation reinforced the quality of the proposed model.

Given enough data to build data-driven models, their potential lies in the way they learn and exploit relations between data unconstrained by functional or parametric assumptions and choices. And, beyond that, the use of these models to reproduce a hydrologist's way to identify rainfall-runoff events is just one of many potential applications.

## 1 Introduction

Discharge time series are essential for various activities in Hydrology and Water resources management. In the words of Chow (1988) "[…] the hydrograph is an integral expression of the physiographic and climatic characteristics that govern the relations between rainfall and runoff of a particular drainage basin". Discharge time series are a fundamental component of hydrological learning and prediction since they i) are relatively easy-to-obtain, available in high quality and from widespread and long-existing observation networks; ii) carry robust and integral information about the catchment state; and iii) are an important target quantity for hydrological prediction and decision-making.

Beyond their value to provide long-term averages aiding water balance considerations, the information they contain about limited periods of elevated discharge can be exploited for baseflow separation, water power planning, sizing of reservoirs and retention ponds, design of hydraulic structures such as bridges, dams or urban storm drainage systems, risk assessment of floods and soil erosion. These periods, essentially characterized by a rising (start), peak and recession (ending) points (Mei and Anagnostou, 2015), will hereafter simply be referred to as "events". They can have many causes (rainfall, snowmelt, upstream reservoir operation, etc.) and equally many characteristic durations, magnitudes and shapes. Interestingly, while for a trained hydrologist with a particular purpose in mind, it is usually straightforward to identify such events in a time series, it is hard to identify them automatically based on a set of rigid criteria. One reason for this is that the set of criteria for discerning events from non-events typically comprises both global and local aspects, i.e., some aspects relate to properties of the entire time series, some to properties in time windows. And to make things worse, the relative importance of these criteria can vary over time, and they strongly depend on user-requirements, hydroclimate and catchment properties.

So why not stick to manual event detection? Its obvious drawbacks are that it is cumbersome, subject to handling errors and hard to reproduce especially when working with long-term data. As a consequence, many methods for objective and automatized event detection have been suggested. The baseflow separation, and consequently the event identification (since the separation allows the identification of the start and end time of the events), has a long history of development. Theoretical and empirical methods for determining baseflow are discussed since 1893 as presented in Hoyt et al. (1936). One of the oldest techniques according to Chow et al. (1988) date back from the early 1930s, with the normal depletion curve from Horton (1933). As stated by Hall (1968), fairly complete discussions of baseflow equations, mathematical derivations and applications were already present in the 1960s. In the last two decades, more recent techniques embracing a multitude of approaches (graphical-, theoretical-, mathematical-, empirical-, physical- and data- based) aim to automate the separation.

Ehret and Zehe (2011) and Seibert et al. (2016) applied a simple discharge threshold approach with partly unsatisfactory results; Merz et al. (2006) introduced an iterative approach for event identification based on the comparison of direct runoff and a threshold. Merz and Blöschl (2009) expanded the concept to analyze runoff coefficients and applied it to a large set of catchments. Blume et al. (2007) developed the "Constant-k" method for baseflow separation employing a gradient-based search for the end of event discharge. Koskelo et al. (2012) presented the physically-based "Sliding Average with Rain Record" - SARR - method for baseflow separation in small watersheds based on precipitation and quickflow response. Mei and

Anagnostou (2015) suggested a physically-based approach for combined event detection and baseflow separation, which provides event start, peak and end times.

While all of these methods have the advantage of being objective and automatable, they suffer from limited generality. The reason is that each of them contains some kind of conceptualized, fixed relation between input and output. Even though this relation can be customized to a particular application by adapting parameters, it remains to a certain degree invariant. In particular, each method requires an invariant set of input data and sometimes it is constrained to a specific scale, which limits its application to specific cases and to where these data are available.

With the rapidly increasing availability of observation data, computer storage and processing power, data-based models have become increasingly popular as an addition or alternative to established modeling approaches in Hydrology and Hydraulics (Solomatine and Osfeld, 2009). According to Solomatine and Osfeld (2008, 2009), they have the advantage of not requiring detailed consideration of physical processes (or any kind of a priori known relation between model input and output); instead, they infer these relations from data, which however requires that there are enough data to learn from. Of course, including a priori known relations among data into models is an advantage as long as we can assure that they really apply. However, when facing undetermined problems, i.e., for cases where system configuration, initial and boundary conditions are not well known, applying these relations may be over-constraining, which may lead to biased and/or overconfident predictions. Predictions based on probabilistic models that learn relations among data directly from the data, with few or no prior assumptions about the nature of these relations, are less bias-prone (because there are no prior assumptions potentially obstructing convergence towards observed mean behavior), and less likely to be overconfident compared to established models (because applying deterministic models is still standard hydrological practice, and they are overconfident in all but the very few cases of perfect models). This applies at least if there is sufficient data to learn from, appropriate binning choices were made (see the related discussion in Section 2.2) and the application remains within the domain of the data that were used for learning.

In the context of data-based modeling in Hydrology, concepts and measures from information theory are becoming increasingly popular to describe and infer relations among data (Liu et al., 2016), quantify uncertainty and evaluate model performance (Chapman, 1986; Liu et al., 2016), estimate information flows (Weijs, 2011; Darscheid, 2017), analyze spatio–temporal variability of precipitation data (Mishra et al., 2009; Brunsell, 2010), describe catchment flow (Pechlivanidis et al., 2016) and measure quantity and quality of information in hydrological models (Nearing and Gupta, 2015).

In this study, we describe and test a data-driven approach for event detection formulated in terms of information theory, showing that its potential goes beyond event classification, since it enables the identification of the drivers of the classification, to choose the most suitable model for an available data set, to quantify minimal data requirements, to automatically reproduce classifications for database generation, to handle any kind of relation between the data. The method is presented in Section 2. In Section 3, we describe two test applications with data from the Dornbirnerach catchment in Austria. We present the results in Section 4 and draw conclusions in Section 5.

## 2 Method description

The core of the information theory method (ITM) is straightforward and generally applicable; its main steps are shown in Fig. 1 and will be explained in the following.

### 2.1 Model hypothesis step

The process starts by selecting the target (what we want to predict) and the predictor data (that potentially contain information about the target). Choosing the predictors constitutes the first and most important model hypothesis, and there are almost no restrictions to this choice: They can be any kind of observational or other data, transformed by the user or not; they can be part of the target data set themselves, e.g., time-lagged or space-shifted; they can even be the output of another model. The second choice and model hypothesis is the mapping between items in the target and the predictor data set, i.e., the relation hypothesis.

It is important for the later construction of conditional histograms that a 1:1 mapping exists between target and predictor data, i.e., one particular value of target is related to one particular value of predictor (in contrast to 1:n or n:m relationships). Often, the mapping relation is established by equality in time.

### 2.2 Model building step

The next step is the first part of model building. It consists of choosing the value range and binning strategy for target and

predictor data. These choices are important as they will frame the estimated multivariate probability mass functions (PMFs) constituting the model and directly influence the statistics we compute from them for evaluation. Generally, these choices are subjective and reflect user-specific requirements and should be made taking into consideration data precision and distribution, size of the available data sets and required resolution of the output. According to Gong et al. (2014), when constructing probability density functions (PDFs) from data via the simple bin-counting method, "[…] too small a bin width may lead to a

histogram that is too rough an approximation of the underlying distribution, while an overly large bin width may result in a histogram that is overly smooth compared to the true PDF.". Gong et al. (2014) also discussed the selection of an optimal bin width by balancing bias and variance of PDF estimation. Pechlivanidis et al. (2016) investigated the effect of bin resolution on the calculation of Shannon Entropy and recommended that bin width should not be less than the precision of the data. Also, while equidistant bins have the advantage of being simple and computationally efficient (Ruddell and Kumar 2009), hybrid

alternatives can overcome weaknesses of conventional binning methods to achieve a better representation of the full range of data (Pechlivanidis et al., 2016).

With the binning strategy fixed, the last part of the model building is to construct a multivariate PMF from all predictors and related target data. The PMF dimension equals the number of predictors plus one (the target), and the way probability mass is distributed within is a direct representation of the nature and strength of the relationship between predictors and target as

contained in the data. Application of this kind of model for a given set of predictor values is straightforward: We simply extract

the related conditional PMF (or PDF) of the target, which, under the assumption of system stationarity, is a probabilistic prediction of the target value.

If the system is non-stationary, i.e., when system properties change with time, the inconsistency between the learning and the prediction situation will result in additional predictive uncertainty. The problems associated with predictions of non-stationary systems apply to all modeling approaches. If a stable trend can be identified, a possible countermeasure is to do learning and prediction on detrended data and then reimpose the trend in a post-processing step.

## 2.3 Model evaluation step

### 2.3.1 Information theory – Measures

In order to evaluate the usefulness of a model, we apply concepts from information theory to select the best predictors (the drivers of the classification) and validate the model. With this in view, this section provides a brief description of the information theory concepts and measures applied in this study. The section is based on Cover and Thomas (2006), which we recommend for a more detailed introduction to the concepts of information theory. Complementary, for specific applications to investigate hydrological data series, we refer the reader to Darscheid (2017).

Entropy can be seen as a measure of the uncertainty about a random variable; it is a measure of the amount of information required on average to describe a random variable (Cover and Thomas, 2006). Let $X$ be a discrete random variable with alphabet $\chi$ and probability mass function $p(x)$, $x \in \chi$. Then, the Shannon Entropy $H(X)$ of a discrete random variable $X$ is defined by

$$H(X) = -\sum_{x \in \chi} p(x) \log_2 p(x) \tag{1}$$

If the logarithm is taken to base two, an intuitive interpretation of Entropy is: "Given prior knowledge of a distribution, how many binary (Yes/No) questions need to be asked on average until a value randomly drawn from this distribution is identified?".

We can describe the Conditional Entropy as the Shannon Entropy of a random variable conditional on the (prior) knowledge of another random variable. The Conditional Entropy $H(X|Y)$ of a pair of discrete random variables $(X, Y)$ is defined as

$$H(X|Y) = -\sum_{y \in Y} p(y) \sum_{x \in \chi} p(x|y) \log_2 p(x|y) \tag{2}$$

The reduction in uncertainty due to another random variable is called the Mutual Information $I(X,Y)$, which is equal to $H(X) - H(X|Y)$. In the study, both measures, Shannon Entropy and Conditional Entropy, are used to quantify the uncertainty of the models (uni- and multivariate probability distributions, respectively). The first is calculated as reference and measures the uncertainty of the target data set. The latter is applied to the probability distributions of the target conditional on predictor(s), and it corroborates to select the more informative predictors, i.e., the ones which lead to the most significant reduction of uncertainty about the target.

It is also possible to compare two probability distributions $p$ and $q$. For measuring the statistical "distance" between the distributions $p$ and $q$, it is common to use Relative Entropy or Kullback-Leibler Divergence $D_{KL}(p||q)$, which is defined as

$$D_{KL}(p||q) = \sum_{x \in \chi} p(x) \log_2 \frac{p(x)}{q(x)} \tag{3}$$

The Kullback-Leibler Divergence is also a measure of the inefficiency of assuming that the distribution is $q$ when the true distribution is $p$ (Cover and Thomas, 2006). The Shannon Entropy $H(p)$ of the true distribution $p$ plus the Kullback-Leibler Divergence $D_{KL}(p||q)$ of $p$ with respect to $q$ is called Cross Entropy $H_{pq}(X||Y)$. In the study, we use these related measures to validate the models and to avoid overfitting, by measuring the additional uncertainty of a model if it is not based on the full data set $p$ but only a sample $q$ thereof.

Note that the uncertainty measured by Eq. (1) to Eq. (3) depends only on event probabilities, not on their values. This is convenient as it allows joint treatment of many different sources and types of data in a single framework.

**2.3.2 Information theory – Model evaluation**

As a benchmark, we can start with the case where no predictor is available, but only the unconditional probability distribution of the target is known. As seen, the associated predictive uncertainty can be measured by the Shannon Entropy $H(X)$ of the distribution (here $X$ indicates the target). If we introduce a predictor and know its value in a particular situation a priori, predictive uncertainty is the entropy of the conditional probability function of the target given the particular predictor value. Conditional Entropy $H(X|Y)$, where $Y$ indicates the predictor(s), is then simply the probability-weighted sum of entropies of all conditional PMFs. Mutual information $I(X, Y)$) is the difference between Shannon Entropy and Conditional Entropy and, like Conditional Entropy, is a generic measure of statistical dependence between variables (Sharma and Mehrotra, 2014), which we can use to compare competing model hypotheses and select the best among them.

Obviously, advantages of setting up data-driven models in the described way are that it involves very few assumptions and it is straightforward to formulate a large number of alternative model hypotheses. However, there is an important aspect we need to consider: from the information inequality, we know that Conditional Entropy is always less than or equal to the Shannon Entropy of the target (Cover and Thomas, 2006). In other words, "information never hurts", and consequently adding more predictors will always either improve or at the least not worsen results. In the extreme, given enough predictors and applying a very refined binning scheme, a model can potentially yield perfect predictions if applied to the learning data set. However, besides the higher computational effort, in this situation, the "curse of dimensionality" (Bellman, 1957) occurs, which "covers various effects and difficulties arising from the increasing number of dimensions in a mathematical space for which only a limited number of data points are available (Darscheid, 2017)". This means that with each predictor added to the model, the dimension of the conditional target-predictor PMF will increase by one, but its volume will increase exponentially. For example, if the target PMF is covered by two bins and each predictor by 100, then a single, double and triple predictor model will consist of 200; 20 000 and 2 000 000 bins, respectively. Clearly, we will need a much larger data set to populate the latter

PMF than the first. This also means that increasing the number of predictors for a fixed number of available data increases the risk of creating an overfitted/non-robust model in the sense that it will become more and more sensitive to the absence or presence of each particular data point. Models overfitted to a particular data set are less likely to produce good results when applied to other data sets than robust models, which capture the essentials of the data relation without getting lost in detail.

We consider this effect with a resampling approach: From the available data set, we take samples of various sizes and construct the model from the sample (see repetition statement regarding $N$ in Fig. 1). Obviously, as the model was built from just a sample, it will not reflect the target-predictor relation as well as a model constructed from the entire data set. It has been shown (Cover and Thomas, 2006; Darscheid, 2017) that the total uncertainty of such an imperfect model is the sum of two components: The Conditional Entropy $H(X|Y)$ of the "perfect" model constructed from all data and the Kullback-Leibler

Divergence $D_{KL}$ between the sample-based and the perfect model. $D_{KL}$ measures the statistical "distance" between the two distributions, in other words, it quantifies the additional uncertainty due to the use of an imperfect model. For a given model (selection of target and predictors), the first summand is independent of the sample size as it is calculated from the full data set, but the second varies: the smaller the sample, the higher $D_{KL}$. Another important aspect of $D_{KL}$ is that for a fixed amount of data, it strongly increases with the dimension of the related PMFs, in other words, it is a measure of the impact of the curse

of dimensionality. In information terms, the sum of Conditional Entropy and Kullback-Leibler Divergence is referred to as Cross Entropy $H_{pq}(X||Y)$. A typical example of Cross Entropy as a function of sample size is, for a single model, shown in Fig. 2.

The curve represents the mean of several repetitions, which were randomly taken with replacement among these repetitions. Note that, comparable to the Monte Carlo cross-validation, the analysis presented in Fig.2 summarizes a large number of

training and testing splits performed repeatedly, and, in addition, also in different split proportions (subsets of various sizes). The difference here is that, in contrast to a standard split where data sets for training and testing are mutually exclusive, we build the model in the training set and apply it in the full data set, where part of the data has not been seen yet, and part has. In other words, we use the training subsets for building the model (a supervised learning approach), and the resulting model is then applied to and evaluated on the full data set. If, on the one hand, the use of the full data set for the application includes

data of the training set, on the other hand, the procedure favors the comparison of the results always with the same model. Thus, the stated procedure allows a robust and holistic analysis, in the sense it works with the mean of $W$ repetition for each subset and compares different sizes of training subset with a unique reference, the model built from the full data set.

Particularly, Fig. 2 shows that for small sample sizes, $D_{KL}$ is the main contributor to total uncertainty, but when the sample approaches the size of the full data set, it disappears, and total uncertainty equals Conditional Entropy. From the shape of the

curve in Fig. 2 we can also infer whether the available data are sufficient to support the model: When $D_{KL}$ approaches zero (Cross Entropy approaches its minimum), this indicates that the model can be robustly estimated from the data, or, in other words, the sample size is enough to represent the full data set. In an objective manner, we can also do a complementary analysis by calculating the ratio $D_{KL}/H(X|Y)$, which is a measure of the relative contribution of $D_{KL}$ to total uncertainty. We can then compare this ratio to a defined tolerance limit (e.g., 5 %) to find the minimally required sample size.

Another application for Fig. 2 is to use these kinds of plots to select the best among competing models with different numbers of predictors. Typically, for small sample sizes, simple models will outperform multi-predictor models as the latter will be hit harder by the curse of dimensionality; but with increasing data availability, this effect will vanish, and models incorporating more sources of information will be rewarded.

In order to reduce the effect of chance when taking random samples, we repeat the described resampling and evaluation procedure many times for each sample size (see repetition statement $W$ in Fig. 1) and take the average of the resulting $D_{KL}$'s and $H_{pq}$'s. Based on these averaged results, we can identify the best model for a set of available data.

The proposed Cross Entropy curve contains a joint visualization of model analysis and model evaluation, and at the same time provide the opportunity of comparing models with different numbers of predictors, being a support tool to decide, for a given

amount of data, which number of predictors is optimal in the sense of avoiding both ignoring available information (by choosing too few predictors) and overfitting (by choosing too many predictors). And, since it incorporates a sort of cross-validation in its construction, one of the advantages of this approach is that it avoids splitting the available data into a training and a testing set. Instead, it makes use of all available data to learn and provides measures of model performance across a range of sample sizes.

## 2.4 Model application step

Once a model has been selected, the ITM application is straightforward: From the multivariate PMF that represents the model, we simply extract the conditional PMF of the target for a given set of predictor values. The model returns a probabilistic representation of the target value. If the model was trained on all available data, and is applied within the domain of these data, the predictions will be unbiased and neither over- nor underconfident. If instead a model using deterministic functions is trained

and applied in the same manner, the resulting single-valued predictions may also be unbiased, but due to their single-value nature will surely be overconfident.

For application in a new time series, if its conditions are outside of the range of the empirical PMF or if they are within the range but have never been observed in the training data set, the predictive distribution of the target (event Yes/No) will be empty and the model will not provide a prediction. Several methods exist to guarantee a model answer, however at the cost of

reduced precision. The solutions range from i) coarse-graining: where the PMF can be rebuilt with fewer, wider bins and an extension of the range until the model provides an answer to the predictive setting, as have been proposed by Darbellay and Vajda (1999), Knuth (2013) and Pechlivanidis et al. (2016); to ii) gap-filling: where the binning is maintained and the empty bins are filled with non-zero values based on a reasonable assumption. Gap-filling approaches comprise adding one counter to each zero-probability bin of the sample histogram, adding a small probability to the sample PDF, smoothing methods such as

Kernel-density smoothing (Blower and Kelsall, 2002; Simonoff, 1996), or Bayesian approaches based on the Dirichlet and Multinomial distribution, or a Maximum-Entropy Method recently suggested by Darscheid et al. (2018). The latter being applied in the present study.

## 3 Design of a test application

In this section, we describe the hydro-climatic properties of the data and the two performed applications. For demonstration purposes, the first test application was developed according to the Section 2, in order to explain which additional predictors we derived from the raw data and related binning and other choices, and present our strategy for model setup, classification and evaluation. For benchmarking purposes, the second application compares the proposed data-driven approach (ITM) with the physically-based approach proposed by Mei and Anagnostou (2015), the characteristic point method (CPM), and applies the holdout method (splitting the data set into training and testing set) for the cross-validation analysis.

### 3.1 Data and site properties

We used quality-controlled hourly discharge and precipitation observations from a 9-year period (Oct/31/1996–Nov/01/2005, 78 912 time steps). Discharge data are from gauge "Hoher Steg", which is located at the outlet of the 113 km² Alpine catchment of the Dornbirnerach River in north-western Austria. Precipitation data are from station "Ebnit" located within the catchment. For the test period, we manually identified hydrological events by visual inspection of the discharge time series. To guide this process, we used a broad event definition, which can be summarized as follows: "An event is a coherent period of elevated discharge compared to the discharge immediately before and after, and/or a coherent period of high discharge compared to the data of the entire time series". We suggest that this is a typical definition if the goal is to identify events for hydrological process studies such as analysis of rainfall-runoff coefficients, baseflow separation, or recession analysis. Based on this definition, we classified each time step of the time series as either being part of an event (value "1") or not (value "0"). Altogether, we identified 177 individual events covering 9092 time steps, which is 11.5 % of the time series. For the available 9-year period, the maximum precipitation is 28.5 mm h$^{-1}$ and the maximum and minimum discharge values are 237.0 m$^3$ s$^{-1}$ and 0.037 m$^3$ s$^{-1}$, respectively. A preliminary analysis revealed that all times with discharge exceeding 15.2 m³ s$^{-1}$ were classified as event, all times with discharge below 0.287 m³ s$^{-1}$ were always classified as non-event.

Both, the input data and the event classification are shown in Fig. 3.

### 3.2 Application I – ITM

### 3.2.1 Predictor data and binning

Since we wanted to build and test a large range of models, we did not only apply the raw observations of discharge and precipitation but also derived new data sets. The target and all predictor data sets with the related binning choices are listed in Table 1; additionally, the predictors are explained in the text below. For reasons of comparability, we applied uniform binning (fixed width interval partitions) to all data used in the study, except for discharge: Here we grouped all values exceeding 15.2 m³ s$^{-1}$ (the threshold beyond which an event occurred for sure) into one group to increase computational efficiency. For each data type, we selected the bin range to cover the range of observed data and chose the number of bins with the objective to maintain the overall shape of the distributions with the least number of bins.

#### 3.2.1.1  Discharge $Q$ [m³ s⁻¹]

This is the discharge as measured at Hoher Steg. In order to predict an event classification at time step $t$, we tested discharge at the same time step as a predictor – $Q(t)$ – and at time steps before – $Q(t-1)$, $Q(t-2)$ – and after – $Q(t+1)$, $Q(t+2)$.

#### 3.2.1.2  Natural logarithm of discharge $\ln Q$ [ln(m³ s⁻¹)]

We also used a log-transform of discharge to evaluate whether this non-linear transformation preserved more information in $Q$ when mapped into the binning scheme than the raw values. Note that the same effect could also be achieved by a logarithmic binning strategy, but as mentioned we decided to maintain the same binning scheme for reasons of comparability. As for $Q$, we applied the log-transform also to time-shifted data.

#### 3.2.1.3  Relative magnitude of discharge $Q_{\mathrm{RM}}$ [-]

This is a local identifier of discharge magnitude at time $t$ in relation to its neighbors within a time window. For each time-step, we normalized discharge into the range [0, 1] using Eq. (4), where $Q_{\max}$ is the largest value of $Q$ within the window and $Q_{\min}$ is the smallest.

$$Q_{\mathrm{RM}} = \frac{Q(t) - Q_{\min}}{Q_{\max} - Q_{\min}} \tag{4}$$

A value of $Q_{\mathrm{RM}} = 0$ indicates that $Q(t)$ is the smallest discharge within the analyzed window, a value of $Q_{\mathrm{RM}} = 1$ indicates that it is the largest. We calculated these values for many window sizes and for windows with the time step under consideration

in the center ($Q_{\mathrm{RMC}}$), at the right end ($Q_{\mathrm{RMR}}$) and at the left end ($Q_{\mathrm{RML}}$) of the window. The best results were obtained for a time-centered window of 65 hours. For further details see Section 3.2.2.

#### 3.2.1.4  Slope of discharge $Q_{\mathrm{slope}}$ [m³ s⁻¹ h⁻¹]

This is the local inclination of the hydrograph. This predictor was created to take into consideration the rate and direction of discharge changes. We calculated both the slope from the previous to the current time step applying Eq. (5) and the slope from

the current to the next time step applying Eq. (6). Positive values always indicate rising discharge.

$$Q_{\mathrm{slope}_{\mathrm{before}}} = \frac{Q(t) - Q(t-1)}{t - (t-1)} \tag{5}$$

$$Q_{\mathrm{slope}_{\mathrm{after}}} = \frac{Q(t+1) - Q(t)}{(t+1) - t} \tag{6}$$

#### 3.2.1.5  Precipitation $P$ [mm h⁻¹]

This is the precipitation as measured at Ebnit.

### 3.2.1.6 Model-based event probability $e_\mathrm{p}$ [-]

In general, information about a target of interest can be encoded in related data such as the predictors introduced above, but it can also be encoded in the ordering of data. This is the case if the processes that are shaping the target exhibit some kind of temporal memory or spatial coherence. For example, the chance of a particular time step to be classified as being part of an event increases if the discharge is on the rise, and it declines if the discharge declines. We can incorporate this information by adding to the predictors discharge from increasingly distant time steps, but this comes at the price of a rapidly increasing impact of the curse of dimensionality. To mitigate this effect, we can use sequential or recursive modeling approaches: In a first step, we build a model using a set of predictors and apply it to predict the target. In a next step, we use this prediction as a new, model-derived predictor, combine it with other predictors in a second model, use it to make a second prediction of the target and so forth. Each time we map information from the multi-dimensional set of predictors onto the 1-dimensional model output, we compress data and reduce dimensionality while hoping to preserve most of the information contained in the predictors. Of course, if we apply such a recursive scheme and want to avoid iterations, we need to avoid circular references, i.e., the output of the first model must no depend on the output of the second. In our application, we assured this by using the output from the first model at time step $t-1$ as a predictor in the second model to make a prediction at time step $t$. Comparable to a Markov model, this kind of predictor helps the model to better stick to a classification after a transition from event to no-event or vice versa.

### 3.2.2 Selecting the optimal window size for the $Q_\mathrm{RM}$ predictor

To select the most informative window size when using relative magnitude of discharge as a predictor, we calculated Conditional Entropy of the target given discharge and the $Q_\mathrm{RMC}$, $Q_\mathrm{RML}$ and $Q_\mathrm{RMR}$ predictors for a range of window sizes on the full data set. The definition of the window sizes for the different window types and the Conditional Entropies are shown in Fig. 4.

The best (lowest) value of Conditional Entropy was obtained for a time-centered window ($Q_\mathrm{RMC}$) with $2 \cdot 32 + 1 = 65\,\mathrm{h}$ of a total width. We used this value for all further analyses.

### 3.2.3 Model classification, selection and evaluation

#### 3.2.3.1 Model classification

All the models we set up and tested in this study can be assigned to one of three distinct groups. The groups distinguish both typical situations of data availability and the use of recursive and non-recursive modeling approaches. Models in the *Q-based group* apply exclusively discharge-based predictor(s). For models in the *P-based group*, we assumed that in addition to discharge, precipitation data are also available. This distinction was made because in the literature exist two main groups of event detection methods: One relying solely on discharge data, the other using precipitation data additionally. Finally, models in the *Model-based group* all apply a 2-step recursive approach as discussed in Section 3.2.1.6. In this case, the first model is

always from the $Q$- or $P$-based group. Later, event predictions at time step $t-1$ of the first model application are then, together with additional predictors from the $Q$- or $P$-based group, used as a predictor in the second model.

#### 3.2.3.2 Model selection

In order to streamline the model evaluation process, we applied an approach of supervised model selection and gradually increasing model complexity: We started by setting up and testing all possible 1-predictor models in the $Q$- and $P$-based group. From these, we selected the best performing model and combined it with each remaining predictor into a set of 2-predictor models. The best performing 2-predictor model was then expanded to a set of 3-predictor models using each remaining predictor and so forth. For the Model-based group, the strategy was to take the best-performing models from both the $Q$- and the $P$-based group as the first model and then combine it with an additional predictor. In the end, we stopped at 4-predictor models, as beyond the uncertainty contribution due to limited sample size became dominant.

#### 3.2.3.3 Model evaluation

Among models with the same number of predictors, we compared model performance via the Conditional Entropy (target given the predictors), calculated from the full data set. However, when comparing models with different numbers of predictors, the influence of the curse of dimensionality needs to be taken into account. To this end, we calculated sample-based Cross Entropy and Kullback-Leibler Divergence as described in Section 2.3.2 for samples of size fifty up to the size of the full data set, using the following sizes [50; 100; 500; 1000; 1500; 2000; 2500; 5000; 7500; 10 000; 15 000; 20 000; 30 000; 40 000; 50 000; 60 000; 70 000; 78 912]. To eliminate effects of chance, we repeated the resampling 500 times for each sample size and took their averages. In Appendix A, the resampling strategy and the choice of repetitions are discussed in more detail.

### 3.3 Application II – ITM and CPM comparison

The second application aims to compare the performances of ITM and another automatic event identification method in a more familiar perspective. The predictions were performed in a new data set, and, as a measure of diagnostic, concepts from the receiver operating characteristic (ROC) curve quantified the hits and misses of the predictions of both models according to a time series of user-classified events (considered the true value). More about the ROC analysis can be found in Fawcett (2005). For the comparison, the characteristic point method (CPM) was chosen because, in contrast with the data-driven ITM, it is a physically-based approach for event identification, which is applicable and recommended to the characteristics of the available data set (hourly time scale data on catchment precipitation and discharge) and open source. The essence of the method is to characterize flow events with three points (start, peak(s) and end of the event) and then associate the event to a corresponding rainfall event (Mei and Anagnostou, 2015). For the event identification, a baseflow separation is previously needed and proposed by coupling the revised constant k method (Blume et al., 2007) and the recursive digital filter proposed by Eckhardt (2005). More about CPM can be attained in Mei and Anagnostou (2015).

Since the outcome of CPM is dichotomous, such as event and non-event, the probabilistic outcome of ITM must be converted into a binary solution. The binarization was reached in the study by choosing an optimum threshold of the probabilistic prediction ($p_{\text{threshold}}$), where all time steps with probabilities equal or greater than it were classified as being part of an event. The objective function of the optimization was based on the ROC curve and sought to minimize the distance to the top-left corner of the ROC curve, i.e., the Euclidean distance between the true positive rate ($R_{\text{TP}}$, proportion of events correctly identified in relation to the total of true events) and false positive rate ($R_{\text{FP}}$, proportion of false events in relation to the total of true non-events) to the perfect model (where $R_{\text{TP}_{\text{perfect}}} = 1$ and $R_{\text{FP}_{\text{perfect}}} = 0$), as expressed in the Eq. (7). A discussion about the cut-off values of ROC curve can be found in Habibzadeh et al. (2016).

$$\min \sqrt{(1 - R_{\text{TP}})^2 + (0 - R_{\text{FP}})^2} \tag{7}$$

Even though the physically-based CPM method theoretically does not require a calibration step, for avoiding misleading comparison, the parameter $R_{\text{nc}}$ (rate of no-change, used to quantify null-change ratio in recession coefficient k) were optimized by Eq. (7). $R_{\text{TP}}$, $R_{\text{FP}} \in [0,1]$ and are calculated as a function of the optimized parameter $p_{\text{threshold}}$ (for ITM) and $R_{\text{nc}}$ (for CPM).

Due to the $p_{\text{threshold}}$ and $R_{\text{nc}}$ optimization and to enable the cross-validation of the models in a new data set, the available data were divided into training and testing sets. And, since ITM model requires a minimum data set size to guarantee the model robustness, the holdout split was based on the data requirement of the selected ITM model obtained according to the application I, Section 3.2. Therefore, the training data set was used to build the ITM model of the target and selected predictors and to calibrate the $p_{\text{threshold}}$ (needed for the binarization) and $R_{\text{nc}}$.

After that, the calibrated models (ITM and CPM) were applied in a new data set (testing data set) and measures of quality based on the ROC curve were computed in order to evaluate and compare their performance, such as: i) the true positive rate ($R_{\text{TP}}$), which represents the percentage of event classification hits (counting of events correctly classified by the model, $P_{\text{T}}$, divided by the amount of the true events in the testing data set, $P$); ii) the false positive rate ($R_{\text{FP}}$), which represents the percentage of false events identified by the model (counting of events misclassified by the model, $P_{\text{F}}$, divided by the amount of the true non-events in the testing data set, $N$); iii) the accuracy, which reflects the total proportion of events ($P_{\text{T}}$) and non-events (or true negative, $N_{\text{T}}$) that were correctly predicted by the model; and iv) the distance to the perfect model given by the Eq. (7), which represents the norm between the results obtained by the method and a perfect prediction.

## 4 Results and discussion

### 4.1 Results for application I

#### 4.1.1 Model performance for the full data set

Here we present and discuss the model results when constructed and applied to the complete data set. As we stick to the complete data set, Kullback-Leibler divergence will always be zero, and model performance can be fully expressed by

Conditional Entropy (see Section 3.2.3.3), with the (unconditional) Shannon Entropy of the target data H($e$) = 0.516 bits as an upper limit, which we use as a reference to calculate the relative uncertainty reduction for each model. In Table 2, Conditional Entropies and their relative uncertainty reductions are shown for each $Q$- and $P$-based 1-predictor model.

1-predictor models based on $Q$ and ln$Q$ reduced uncertainty to about 50 % (models #1-10 in Table 2, column 4), with a slight advantage of $Q$ over ln$Q$. Interestingly, both show their best results for time offset $t + 2$, i.e., future discharge is a better predictor for event detection than discharge at the current time step. As we were not sure whether this also applies to 2-predictor models, we decided to test both the $t + 2$ and $t$ predictors of $Q$ and ln$Q$ in the next step. Compared to $Q$ and ln$Q$, relative magnitude of discharge $Q_{\mathrm{RMC}}$ and discharge slope $Q_{\mathrm{slope}}$ performed poorly, and so did P, the only model in the $P$-based group. Most likely this is because for a certain time step, to be part of an event is not so much dependent on precipitation at this particular time step, but rather on the accumulated rainfall in a period preceding it. Despite its poor performance, we decided to use it in higher-order models to see whether it becomes more informative in combination with other predictors.

Based on these considerations and the model selection strategy described in Section 3.2.3.2, we built and evaluated all possible 2-predictor models. The models and results are shown in Table 3.

As could be expected from the information inequality, adding a predictor improved the results, and for some models (#16 and #20), the $t$-predictors outperformed their $t + 2$ counterparts (#17 and #21, respectively). Once more, $Q$ predictors performed slightly better than ln$Q$, such that for all higher-order models, we only used $Q(t)$ and ignored $Q(t + 2)$, ln$Q(t)$ and ln$Q(t + 2)$.

In the $P$-based group, adding any predictor greatly improved results by about 50 %, but not a single P-based model outperformed even the worst from the $Q$-based group.

Finally, from both the $Q$- and $P$-based group, we selected the best model (#16 and #23, respectively) and extended them to 3-predictor models with the remaining predictors. The models and results are shown in Table 4.

Again, for both models, the added predictor improved results considerably, and we used both of them to build a recursive 4-predictor model as described in Section 3.2.3. The new predictor, $e_{\mathrm{p}}(t - 1)$ is simply the probabilistic prediction of a model (#27 or #28, in this case) for time step $t - 1$ of being part of an event, with value range [0, 1]. This means $e_{\mathrm{p}_{\#27}}(t - 1)$ carries the memory from the previous predictions of model #27 (and $e_{\mathrm{p}_{\#28}}(t - 1)$ from model #28, accordingly), and the new 4-predictor models #29 and #30 as shown in Table 5 are simply copies of these models, extended by a memory term: $e_{\mathrm{p}}(t - 1)$. Again, model performance improved, and model #29 was the best among all tested models. Though, so far the effect of sample size was not considered, which might have a strong impact on the model rankings. This is investigated in the next section.

### 4.1.2 Model performance for samples

The sample-based model analysis is computationally expensive, so we restricted these tests to a subset of the models from the previous section. Our selection criteria were to i) include at least one model from each predictor group; ii) include at least one model from each dimension of predictors; and iii) choose the best performing model. Altogether we selected the seven models

shown in Table 6. Please note that despite our selection criteria, we ignored the 1-predictor model using precipitation due to its poor performance.

For these models, we computed the Cross Entropies between the full data set and each sample size $N$ for $W$ repetitions and, in the end, for each sample size $N$, we took the average of the $W$ repetitions. The results are shown in Fig. 5. For comparison, the Cross Entropies between the target data set and samples thereof are also included and labeled as model #0.

In Fig. 5, the Cross Entropies at the right end of the x-axis, where the sample contains the entire data set, equal the Conditional Entropies, as the effect of sample size is zero. However, with decreasing sample size, Cross Entropy grows in a non-linear fashion as $D_{KL}$ starts to grow. If we walk through the space of sample sizes in the opposite direction, i.e., from left to right, we can see that as the samples grow, the rate of change of Cross Entropy decreases, the reason being that the rate of change of $D_{KL}$ decreases, which means that the model learns less and less from new data points. Thus, by visually exploring these "learning curves" of the models we can make two important statements related to the amount of data required to inform a particular model: We can state how large a training data set should be to sufficiently inform a model, and we can compare this size to the size of the actually available data set. If the first is much smaller than the latter, we gain confidence that we have a well-informed, robust model. If not, we know that it may be beneficial to gather more data, and if this is not possible, we should treat model predictions with caution.

As mentioned in Section 2.3.2, besides Fig. 5 informing the amount of data needed to have a robust model (implying that sample size is enough to represent the full data set), it allows comparing competing models with different dimensions and select the optimal number of predictors (taking advantage of the available information and avoiding overfitting). In this sense, in the $P$-based group and for sample sizes smaller 5000, the 1-predictor model #23 performs best, but for larger samples sizes, the 4-predictor model #30 takes the lead. Likewise, in the $Q$-based group and for sample sizes smaller than 2500, the single-predictor model #3 is the best but is outperformed by the 2-predictor model #16 from 2500 until 10 000, which in turn is outperformed by the 4-predictor model #29 from 10 000 to the end. Across all groups, models #3, #16 and #29 form the lower envelope curve in Fig. 5, which means that one of them is always the best model choice, depending on the sample size.

Interestingly, the best performing model for large sample sizes (#29) includes predictors which reflect the definition criteria that guided manual event detection (Section 0): $Q(t)$ and $Q(t+2)$ contain information about the absolute magnitude of discharge, $Q_{RMC}$ expresses the magnitude of discharge relative to its vicinity, and $e_{p_{\#27}}(t-1)$ relates to the requirement of events to be coherent.

We also investigated the contribution of sample-size effects to total uncertainty by analyzing the ratio of $D_{KL}$ and $H(X|Y)$ as described in Section 2.3. As expected, for all models the contribution of sample-size effects to total uncertainty decreases with increasing sample size, but the absolute values and the rate of change strongly differ. For the 1-predictor model #3, $D_{KL}$ contribution is small already for small sample sizes (circa 65 % for sample size equal 50), and it quickly drops to almost zero with increasing sample size. For multi-predictor models such as #29, $D_{KL}$ contribution to uncertainty exceeds that of $H(X|Y)$

by a factor of seven for small samples (circa 700 % for sample size equal 50), and it decreases only slowly with increasing sample size.

In Table 7 (column 5), we show for each model the minimum sample size to keep the $D_{KL}$ contribution below a threshold of 5 %.

As expected, the models with few predictors require only small samples to meet the 5 % requirement (starting from a subset of 12.6 % of the full data set for 1-predictor model to 37.3 % for 2-predictor), but for multi-predictor models such as models #29 and #30, more than 60 000 data points are required (87.6 % and 79.4 % of the full data set, respectively). This happens because the greater the number of predictors, the greater is the number of bins in the model. This means that we need a much larger data set to populate the PMF with the largest number of bins, for example: model #29 has 279 752 bins and requests 7.9 years of data). Considering that the amount of data available in the study is limited, this also means that increasing the number of predictors/bins, the risk of creating an overfitted/non-robust model also increases. Thus, the ratio $D_{KL}/H(X|Y)$ and visual inspection of the curve in Fig. 5 orientate the user when to stop adding new predictors to avoid overfitting. In this fashion, Table 7 shows that each of the models tested meets the 5 % requirement, claiming up to 87.6 % of the available data set (69 102 out of 78 912 data points for model #29), which indicates that all of them are robustly supported by the data. In this case, we can confidently choose the best-performing among them (#29, with uncertainty equal to 0.114 bits) for further use. Interestingly, with this analysis, it was also possible to identify the drivers of the user classification, which, in the case of model #29, were the predictors $Q(t)$, $Q_{RMC}$, $Q(t + 2)$ and $e_p(t - 1)$.

### 4.1.3 Model application

In the previous sections, we developed, compared and validated a range of models to reproduce subjective, manual identification of events in a discharge time series. Given the available data, the best model was a 4-predictor recursive model applying $Q(t)$, $Q_{RMC}$, $Q(t + 2)$ and $e_p(t - 1)$ as predictors and built with the full data set (#29, Table 7). This model reduced the initial predictive uncertainty by 77.8 %, decreasing Conditional Entropy from 0.516 to 0.114 bits. This sounds reasonable, but what do the model predictions actually look like? As an illustration, we applied the model to a subset of the training data, April 26 to June 21, 2001. For this period, the observed discharge, the manual event classification by the user and the model-based prediction of event probability are shown in Fig. 6.

In the period from June 01 to 21, four distinct rainfall-runoff events occurred which were also classified as such by the user. During these events, the model-based predictions for event probability remained consistently high except for some times at the beginning and end of events, or in times of low flow during an event. Obviously, the model here agrees with the user classification, and if we wished to obtain a binary classification from the model, we could get it by introducing an appropriate probability threshold (as further described in Section 4.2).

Things look differently, though, in the period of April 26 to May 10, when snowmelt induced diurnal discharge patterns. During this time, the model identified several periods with reasonable (above 50 %) event probability, but the user classified

none. Arguably, this is a difficult case for both manual and automated classification, as the overall discharge is elevated, but not much, and diurnal events can be distinguished, but are not pronounced. In such cases, both the user-based and the model-based classification are uncertain and may disagree.

To identify snowmelt events or potentially improve the information contained in the precipitation set, other predictors could have been used in the analysis (such as aggregated precipitation, snow depth, air temperature, nitrate concentrations, moving average of discharge, etc.) or the target could have been classified according to it type (rainfall, snowmelt, upstream reservoir operation, etc.), instead of having a dichotomous outcome, such as event and non-event. The choice of target and potential predictors occurs according to user interest and data availability.

Another point that may be of interest to the user is the improvement of the consistency of event duration. This can be reached by selection of predictors or through a post-processing step. As previously discussed in Section 3.2.1.6, by applying a recursive predictor $e_\mathrm{p}(t-1)$ a memory effect is incorporated into the model, bringing some inertia for the transition from event to no-event or vice versa. If it is the user interest, the memory effect could be further enhanced by adding more recursive predictors, such as $e_\mathrm{p}(t-2), e_\mathrm{p}(t-3)$ and so on. An alternative option to clear very-short discontinuous time steps or very-short events would be to increase event coherence in a post-processing step with an autoregressive model, with model parameters found by maximizing agreement with the observed events.

Finally, in contrast to the evaluation approach presented, where the subsets are compared to the full data set (subset data plus data not seen during training), the next section will present the evaluation of ITM and CPM applied for mutually exclusives training and testing sets.

## 4.2 Results for application II

Section 4.1 showed that, for the full data set, the best model was the recursive one with $Q(t)$, $Q_\mathrm{RMC}$, $Q(t+2)$ and $e_\mathrm{p}(t-1)$ as the drivers of the user classification (model #29, Table 7), which could be robustly built with a sample size of 69 102. Thus, to assure its robustness for the second application, since we are creating a new PMF based only on the training set, the split of the data (discharge, precipitation and user event classification) divided the 78 912 time steps in two periods composed by: i) 87.6 % of the full data set (69 102 time steps) forming the training data set (from Oct/31/1996 1:00 to Sep/18/2004 6:00); and the remaining 12.4 % (9810 time steps) the testing data set (from Sep/18/2004 7:00 to Nov/01/2005 00:00). The characteristics of the user event classification data set, used as the true classification for accounting the hits and misses of ITM and CPM, is presented in Table 8.

For model training, input data from both models, ITM and CPM, were smoothed. First, a 24-hour moving average was applied to the discharge of CPM (this was recommended by the first author of the method in personal communication) and, to avoid misleading comparison, it was then applied to the probabilities of ITM right before the binarization. The smoothing improved the results of both models and worked as a post-processing filter which removed some noise (very-short-duration events) and

attenuated effects from snow melting. Note that this is a feature of our training data set, and therefore it is not necessarily applicable to other similar problems and neither is a required step.

Following the data smoothing, we proceeded with the optimization of parameters: threshold for the probability output of ITM and rate of no-change for CPM (Section 3.3). The results of the two models also improved with the optimization performed.

The optimum parameters obtained were $p_{\text{threshold}} = 0.26$ and $R_{\text{nc}} = -6.6$. For these values, the final distances in the training data set given by Eq. (7) were: 0.05 and 0.23 for ITM and CPM, respectively.

After the model training, the calibrated models were applied to the testing data set to predict binary events. The event predictions were then compared to the true classification (Table 8, training line) and their hits and misses were calculated in order to evaluate and compare their performance. The results are compiled in Table 9.

The quality parameters presented in Table 9 show that the ITM true positive rate equals 97.5 %, i.e., 13.0 % higher than CPM $R_{\text{TP}}$). On the other hand, CPM false negative rate is equal to 9.9 %, while ITM $R_{\text{FP}}$ is equal to 12.6 % (2.7 % higher). These results indicate that ITM is more likely to predict events than the CPM but at the cost of increasing the false positive rate. Combining these two rates into a single success criterion according to Eq. (7) showed ITM to be slightly superior to CPM. Considering only the hits of the models, both models performed similarly, reaching almost 90 % of accuracy, with CPM being

slightly better than ITM. However, it should be emphasized that although the accuracy of the model gives a good notion of the model hits, it was not used as a criterion for success because it is a myopic criterion for the false event classifications. False positives are essential in the context of event prediction, since most of the data are non-events (88.2 % of the training data set, Table 8), and a blind classification of all time steps as being non-event, for example, would overcome the accuracy obtained by both models (90.4 % of the testing data set, Table 8), even though it is not a useful model.

As an illustration, in the context of the binary analysis, the observed discharge, the true event classification (manually made by an expert), ITM predicted events and CPM predicted events are shown in Fig. 7 for a subset of the testing data, April 22 to August 23, 2005.

For the analyzed subset, nine distinct rainfall-runoff events occurred and were identified as such by ITM and CPM. However, differently from the true identification, both models grouped some of these events (July 20, August 07 and 16) in events with

longer duration. False events were also observed in both models, where three false events were identified by ITM (July 5, July 7 and July 26) and two (but contemplating the same period as ITM) by CPM. It should be noted that they are false in relation to the user classification; however, we can not exclude the possibility of false classification by the visual inspection process. A further criticism is that the holdout cross-validation involves a single run, which is not as robust as multiple runs. Nevertheless, the way that the split was proposed recognizes the logical order of obtaining the data. Thus, despite the

subjectivity of event selection by a user and the application of a simplified method of cross-validation, it is possible to conclude that, overall, ITM and CPM behaved similarly and provided reasonable predictions, as seen numerically in Table 9 and qualitatively through Fig. 7.

An interesting conclusion is that ITM was able to overcome CPM requiring only discharge data and a training data set of classified events (also based on the discharge set), while CPM demanded precipitation, catchment area and discharge as inputs.

It is important to note that CPM can be modified to be used without precipitation data, however in our case it resulted in a considerably higher false positive rate, since the rainfall event-related filters cannot be applied. On the other hand, since CPM is a physically-based approach, it does not require a training data set with identified events (although the optimization in the calibration step has representatively improved its results) and there are no limitations in terms of data set size, which eliminates the robustness analysis, being then a method more easily implemented for binary classification. The binarization of the ITM predictions and parameter optimization in CPM are not included in the original methods, however, they were essential adaptations to allow a fair comparison of the models. Finally, the suitability or not of the existing event detection techniques depends mainly on the user's interest and the data available for application.

## 5 Summary and conclusions

Typically, it is easy to manually identify rainfall-runoff events, due to the high discriminative and integrative power of the brain-eye system. However, this is i) cumbersome for long time series; ii) subject to handling errors; and iii) hard to reproduce since it dependents on acuity and knowledge of the event identifier. To mitigate these issues, this study has proposed an information theory approach to learn from data, and to choose the best predictors, via uncertainty reduction, to create predictive models for automatically identifying rainfall-runoff events in discharge time series.

The method was established in four main steps: model hypothesis, building, evaluation and application. Each association of predictor(s) to the target is equivalent to formulating a model hypothesis. For the model building, non-parametric models constructed discrete distributions via bin-counting, requiring at least a discharge time series and a training data set containing a Yes/No event identification as target. In the evaluation step, we used Shannon Entropy and Conditional Entropy to select the more informative predictors, and Kullback-Leibler Divergence and Cross Entropy to analyze the model in terms of overfitting and curse of dimensionality. Finally, the best model was applied to its original data set to compare the predictability of the events. For benchmarking purpose, a holdout cross-validation a comparison of the proposed data-driven method with an alternative physically-based approach was performed.

The approach was applied to discharge and precipitation data from the Dornbirnerach catchment in Austria. In this case study, 30 models based on 14 predictors were built and tested. Among these, seven predictive models with a number of predictors varying from one to four were selected. Interestingly, across these models, the three best performing ones were obtained using only discharge-based predictors. The overall best model was a recursive one applying four predictors: discharge from two different time steps, the relative magnitude of discharge compared to all discharge values in a surrounding 65-hour time window and event predictions from the previous time step. Applying the best model, the uncertainty about event classification was reduced by 77.8 %, decreasing Conditional Entropy from 0.516 to 0.114 bits. Since the Conditional Entropy reduction of the models with precipitation was not higher than the ones exclusively based on discharge information, it was possible to infer that: i) the information coming from precipitation was likely already contained in the discharge data series; and ii) the event classification is not so much dependent on precipitation at a particular time step, but rather on the accumulated rainfall in the

period preceding it. Furthermore, precipitation data are often not available for analysis, which makes the model exclusively based on discharge data even more attractive.

Further analysis using Cross Entropy and Kullback-Leibler Divergence showed that the robustness of a model quickly dropped with the number of predictors used (an effect known as the curse of dimensionality) and that the relation between number of predictors and sample size was crucial to avoid overfitting. Thus, the model choice is a tradeoff between predictive power and robustness given the available data. For our case, the minimum amount of data to build a robust model varied from 9952 data points (1-predictor model with 0.260 bits of uncertainty) to 69 102 data points (4-predictor model with 0.114 bits of uncertainty). Complementarily, the quality of the model was verified in a more traditional way, by a cross-validation analysis (where the model was built and validated in a training data set), and a comparative investigation between our data-driven approach and a physically-based model. As a result, in general, both models presented reasonable predictions and reached similar quality parameters, with almost 90 % of accuracy. In the end, the comparative analysis and cross-validation reinforced the quality of the method, previously validated in terms of robustness using measures from information theory.

In the end, the data-driven approach based on information theory is a consolidation of descriptive and experimental investigations, since it allows to describe the drivers of the model through predictors and it investigates the similarity of the model hypothesis with respect to the true classification. In summary, it presents advantages such as: i) It is a general method that involves a minimum of additional assumptions or parametrizations; ii) Due to its non-parametric approach, it preserves as much as possible the full information of the data, which might get lost when expressing the data-relations by functional relationships; iii) It obtains data-relations from the data itself; iv) It is flexible in terms of data requirement and model building; v) It allows to measure the amount of uncertainty reduction via predictors; vi) It is a direct way to account for uncertainty; vii) It permits explicitly comparing information from various sources in a single currency, "bit"; viii) It allows to quantify minimal data requirements; ix) It enables to investigate the curse of dimensionality; x) It is a way of understanding the drivers (predictors) of the model (also useful in machine learning, for example); xi) It permits to choose the most suitable model for an available data set; and xii) The predictions are probabilistic, which compared to a binary classification additionally provides a measure of the confidence of the classification.

Although the procedure was employed to identify events from a discharge time series, which for our case were mainly triggered by rainfall and snowmelt, the method can be applied to reproduce user classification of any kind of event (rainfall, snowmelt, upstream reservoir operation, etc.) and even identify them separately. Moreover, one of the strengths of the data-based approach is that it potentially accepts any data to serve as predictors, and it can handle any kind of relation between the predictor(s) and the target. Thus, the proposed approach can be conveniently adapted to another practical application.

## 6 Data availability

The Event Detection program, containing the functions to develop multivariate histograms and calculate information theory measures, is published alongside this manuscript via GitHub https://github.com/KIT-HYD/EventDetection. The repository

also includes scripts to exemplify the use of the functions and the data set of identified event, discharge and precipitation time series from the Dornbirnerach catchment in Austria used in the case study.

## 7 Author contribution

UE and PD developed the model program (calculation of information theory measures, multivariate histograms operations, event detection) and developed a method to avoid infinitely large values of $D_{KL}$ (Darscheid et al., 2018)). ST performed the simulations, cross-validation, parameter optimization, comparative analysis to a second model and the justification of the number of repetitions of the resampling stage. ST and UE directly contributed to the design of the method and test application, to the analysis of the performed simulations, and wrote the manuscript.

## 8 Acknowledgments

The first and third author acknowledge support by Deutsche Forschungsgemeinschaft DFG and Open Access Publishing Fund of Karlsruhe Institute of Technology (KIT). We thank Clemens Mathis from Wasserwirtschaft Vorarlberg, Austria for providing the case study data.

## Appendix A: Resampling strategy and number of repetitions

In the study, samples of size $N$ from the data set were obtained through bootstrapping, i.e., they were taken randomly, but continuously in time, with replacement among the $W$ repetitions. For each sample size, we repeated draws $W$ times and took the average Cross Entropy and $D_{KL}$ to eliminate effects of chance (see repetition statements $N$ and $W$ in Fig. 1).

Thus, in order to find the value of $W$ which balances statistical accuracy and computational efforts, we did a dispersion analysis through calculating the Shannon Entropy (as a measure of dispersion) of the Cross Entropy distribution of the (unconditional) target model (model #0 in Table 7). Sixty one bins ranging from 0 to 6 in steps of 0.1 bits were used, which contemplates the range of all possible Cross Entropy values among the tested pairs of $N$ and $W$. Fig. A1 presents the Shannon Entropy applied as a dispersion parameter to analyze the effect of the number of repetitions $W$ for different sample sizes $N$.

Considering the graph in Fig. A1, in general, the behavior of the Shannon Entropy among the repetitions is similar for each sample size analyzed, indicating the dispersion of the results according to the number of repetitions does not vary too much, i.e., the bins are similarly filled. However, it is possible to see that, as the sample size increases, the Shannon Entropy for the different number of repetitions approaches that for the 50 000 repetitions. For sample sizes up to 7500, the bars from 50, 100 and 300 repetitions present some peaks and troughs, indicating some dispersion in filling the bins. Thus, in this case study, the minimum of 500 repetitions was assumed as a reasonable number of repetitions to compute the mean of the Cross Entropy in

the sample size investigation. This number of repetitions was also validated considering the smoothness and logical behavior of the curves obtained during the data size validation and curse of dimensionality analyzes (Fig. 5 in Section 4.1.2).

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

**Table 1: Target and predictors – Characterization and binning strategy.**

| Target ($X$) | Symbol | Unit | Bins* [start:end] | Number of bins |
|---|---|---|---|---|
| User-based event classification at time $t$ | e | [-] | [0:1] | 2 |

| Predictors ($Y$) | Symbol | Unit | Bins* [start:step:end] & [start:end] | Number of bins |
|---|---|---|---|---|
| Discharge at times $t$-2, $t$-1, $t$, $t$+1, $t$+2 | $Q(t\text{-}2), Q(t\text{-}1), Q(t),$ $Q(t+1), Q(t+2)$ | [m³ s⁻¹] | [0:0.5:16] & [16:end] | 34 |
| Natural logarithm of discharge at times $t$-2, $t$-1, $t$, $t$+1, $t$+2 | $\ln Q(t\text{-}2), \ln Q(t\text{-}1),$ $\ln Q(t), \ln Q(t+1),$ $\ln Q(t+2)$ | [$ln$(m³ s⁻¹)] | [-3.5:0.2:2.9] & [2.9:end] | 34 |
| Relative Magnitude of discharge in time windows (centered, left-ended, right-ended) | $Q_{RMC}, Q_{RML}, Q_{RMR}$ | [-] | [0:0.1:1] | 11 |
| Discharge slope between time steps $t$-1 and $t$ & time steps $t$ and $t$+1 | $Q_{slope_{before}}$ $Q_{slope_{after}}$ | [m³ s⁻¹ h⁻¹] | [-50:5:90] | 29 |
| Precipitation at time $t$ | $P$ | [mm h⁻¹] | [0:1:30] | 31 |
| Model-based event probability at time step $t$-1 | $e_p(t\text{-}1)$ | [-] | [0:0.1:1] | 11 |

* Bins identified by their central values [leftmost center value : step : rightmost center value]

**Table 2: Conditional Entropy and relative uncertainty reduction of 1-predictor models.**

| # | Predictive model $(X\|Y)$ | $H(X\|Y)$ [bit] | $H(X\|Y)/H(X)$* |
|---|---|---|---|
| | *Q*-based group | | |
| 1 | $e \mid Q(t-2)$ | 0.269 | 52.1 % |
| 2 | $e \mid Q(t-1)$ | 0.264 | 51.3 % |
| 3 | $e \mid Q(t)$ | 0.260 | 50.3 % |
| 4 | $e \mid Q(t+1)$ | 0.255 | 49.4 % |
| 5 | $e \mid Q(t+2)$ | 0.250 | 48.6 % |
| 6 | $e \mid \ln Q(t-2)$ | 0.269 | 52.2 % |
| 7 | $e \mid \ln Q(t-1)$ | 0.265 | 51.3 % |
| 8 | $e \mid \ln Q(t)$ | 0.260 | 50.4 % |
| 9 | $e \mid \ln Q(t+1)$ | 0.255 | 49.4 % |
| 10 | $e \mid \ln Q(t+2)$ | 0.251 | 48.6 % |
| 11 | $e \mid Q_{\mathrm{RMC}}$ | 0.505 | 97.9 % |
| 12 | $e \mid Q_{\mathrm{slope_{before}}}$ | 0.473 | 91.8 % |
| 13 | $e \mid Q_{\mathrm{slope_{after}}}$ | 0.473 | 91.8 % |
| | *P*-based group | | |
| 14 | $e \mid P$ | 0.472 | 91.6 % |

* $H(X) = H(e) = 0.516$ bits

**Table 3: Conditional Entropy and relative uncertainty reduction of 2-predictor models.**

| # | Predictive model $(X\|Y)$ | $H(X\|Y)$ [bit] | $H(X\|Y)/H(X)$* |
|---|---|---|---|
| | *Q*-based group | | |
| 15 | $e \mid Q(t+2), Q(t)$ | 0.226 | 43.9 % |
| 16 | $e \mid Q(t), Q_{\mathrm{RMC}}$ | 0.182 | 35.3 % |
| 17 | $e \mid Q(t+2), Q_{\mathrm{RMC}}$ | 0.191 | 37.1 % |
| 18 | $e \mid Q(t), Q_{\mathrm{slope_{after}}}$ | 0.254 | 49.3 % |
| 19 | $e \mid \ln Q(t+2), \ln Q(t)$ | 0.233 | 45.1 % |
| 20 | $e \mid \ln Q(t), Q_{\mathrm{RMC}}$ | 0.185 | 35.8 % |
| 21 | $e \mid \ln Q(t+2), Q_{\mathrm{RMC}}$ | 0.194 | 37.5 % |
| 22 | $e \mid \ln Q(t), Q_{\mathrm{slope_{after}}}$ | 0.254 | 49.3 % |
| | *P*-based group | | |
| 23 | $e \mid Q(t), P$ | 0.248 | 48.2 % |
| 24 | $e \mid Q(t+2), P$ | 0.247 | 48.0 % |
| 25 | $e \mid \ln Q(t), P$ | 0.249 | 48.2 % |
| 26 | $e \mid \ln Q(t+2), P$ | 0.249 | 48.2 % |

* $H(X) = H(e) = 0.516$ bits

**Table 4: Conditional Entropy and relative uncertainty reduction of 3-predictor models.**

| # | Predictive model $(X\|Y)$ | $H(X\|Y)$ [bit] | $H(X\|Y)/H(X)$* |
|---|---|---|---|
| | $Q$-based group | | |
| 27 | $e \mid Q(t), Q_{\mathrm{RMC}}, Q(t+2)$ | 0.144 | 28.0 % |
| | $P$-based group | | |
| 28 | $e \mid Q(t), P, Q_{\mathrm{RMC}}$ | 0.167 | 32.5 % |

* $H(X) = H(e) = 0.516$ bits

**Table 5: Conditional Entropy and relative uncertainty reduction of recursive 4-predictor models.**

| # | Predictive model $(X\|Y)$ | $\mathbf{H}(X\|Y)$ [bit] | $\mathbf{H}(X\|Y)/\mathbf{H}(X)$* |
|---|---|---|---|
| | Model-based group | | |
| 29 | $e \mid Q(t), Q_{\mathrm{RMC}}, Q(t+2), e_{\mathrm{p}_{\#27}}(t-1)$ | 0.114 | 22.2 % |
| 30 | $e \mid Q(t), P, Q_{\mathrm{RMC}}, e_{\mathrm{p}_{\#28}}(t-1)$ | 0.142 | 27.6 % |

* H$(X)$ = H$(e)$ = 0.516 bits

**Table 6: Models selected for sample-based tests.**

| Model group | 1 predictor | 2 predictors | 3 predictors | 4 predictors |
|---|---|---|---|---|
| $Q$-based group* | $Q(t)$ <br> #3 | $Q(t), Q_{\text{RMC}}$ <br> #16 | $Q(t), Q_{\text{RMC}}, Q(t+2)$ <br> #27 | - |
| $P$-based group** | - | $Q(t), P$ <br> #23 | $Q(t), P, Q_{\text{RMC}}$ <br> #28 | - |
| Model-based group with $Q$-based predictors* | - | - | - | $Q(t), Q_{\text{RMC}}, Q(t+2), e_{\text{p}_{\#27}}(t-1)$ <br> #29 |
| Model-based group with $P$-based predictors** | - | - | - | $Q(t), P, Q_{\text{RMC}}, e_{\text{p}_{\#28}}(t-1)$ <br> #30 |

\* Models which apply exclusively discharge-based predictor(s)

\*\* Models which apply discharge- and precipitation-based predictor(s)

**Table 7: Application I – Curse of dimensionality and data size validation for models in Table 6.**

| # | Predictive model | $H(X)$ [bit] | $H(X)/H(X)^*$ | Sample size where $D_{KL}/H(X) \leq 5\%$ and % of the full data set** | Sample size [a] | Number of bins |
|---|---|---|---|---|---|---|
| 0 | $e$ | 0.516 | 100 % | $\geq 4398$ (5.6 %) | 0.5 | 2 |

| # | Predictive model | $H(X\|Y)$ [bit] | $H(X\|Y)/H(X)^*$ | Sample size where $D_{KL}/H(X\|Y) \leq 5\%$ and % of the full data set** | Sample size [a] | Number of bins |
|---|---|---|---|---|---|---|
| 3 | $e \mid Q(t)$ | 0.260 | 50.4 % | $\geq 9952$ (12.6 %) | 1.1 | 68 |
| 16 | $e \mid Q(t), Q_{\mathrm{RMC}}$ | 0.182 | 35.3 % | $\geq 29\,460$ (37.3 %) | 3.4 | 748 |
| 23 | $e \mid Q(t), P$ | 0.248 | 48.2 % | $\geq 18\,880$ (23.9 %) | 2.2 | 2108 |
| 27 | $e \mid Q(t), Q_{\mathrm{RMC}}, Q(t+2)$ | 0.144 | 28.0 % | $\geq 60\,178$ (76.3 %) | 6.9 | 25\,432 |
| 28 | $e \mid Q(t), P, Q_{\mathrm{RMC}}$ | 0.167 | 32.5 % | $\geq 50\,377$ (63.8 %) | 5.8 | 23\,188 |
| 29 | $e \mid Q(t), Q_{\mathrm{RMC}}, Q(t+2), e_{\mathrm{p}_{\#27}}(t-1)$ | 0.114 | 22.2 % | $\geq 69\,102$ (87.6 %) | 7.9 | 279\,752 |
| 30 | $e \mid Q(t), P, Q_{\mathrm{RMC}}, e_{\mathrm{p}_{\#28}}(t-1)$ | 0.142 | 27.6 % | $\geq 62\,667$ (79.4 %) | 7.2 | 255\,068 |

\* $H(X) = H(e) = 0.516$ bits
\*\* size of the full data set: 78 912 data points (9 years)

**Table 8: Cross-validation data set – Characteristics of the user event classification set.**

| Data set | Time steps classified as positive events ($P$) | Time steps classified as non-events ($N$) | Percentage of events ($P/T$) | Percentage of non-events ($N/T$) | Total ($T$) |
|---|---|---|---|---|---|
| Training | 8150 | 60 952 | 11.8 % | 88.2 % | 69 102 |
| Testing | 942 | 8868 | 9.6 % | 90.4 % | 9810 |
| Sum | 9092 | 69 820 | 11.5 % | 88.5 % | 78 912 |

**Table 9: Application II – ITM and CPM performance.**

| Event detection method | True Positive ($P_T$) | $R_{TP}$ ($P_T/P^*$) | False Positive ($P_F$) | $R_{FP}$ ($P_F/N^*$) | Accuracy % (($P_T + N_T^{**}$)/($P^*+N^*$)) | Eq. (7) distance*** |
|---|---|---|---|---|---|---|
| ITM | 918 | 97.5 % | 1113 | 12.6 % | 88.4 % | 0.13 |
| CPM | 796 | 84.5 % | 877 | 9.9 % | 89.6 % | 0.18 |

\* $P = 942$, $N = 8868$ (Table 8)
\*\* $N_T = N - P_F$
\*\* Distance to the perfect model of the ROC curve

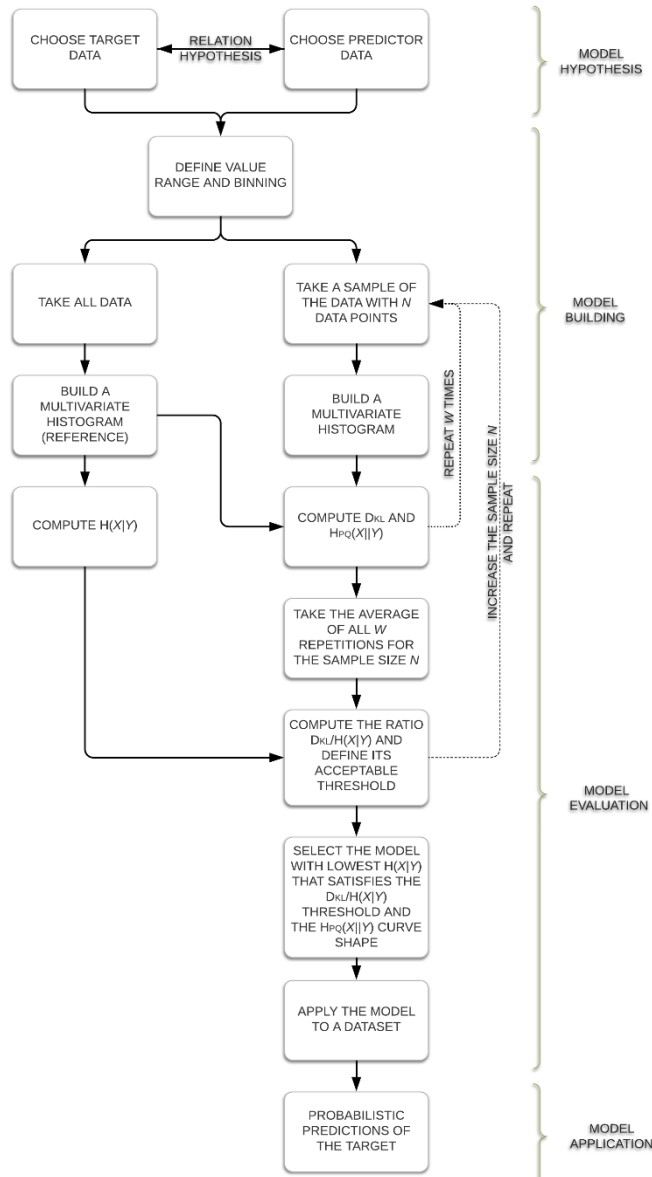

**Figure 1: Main steps of the ITM.**

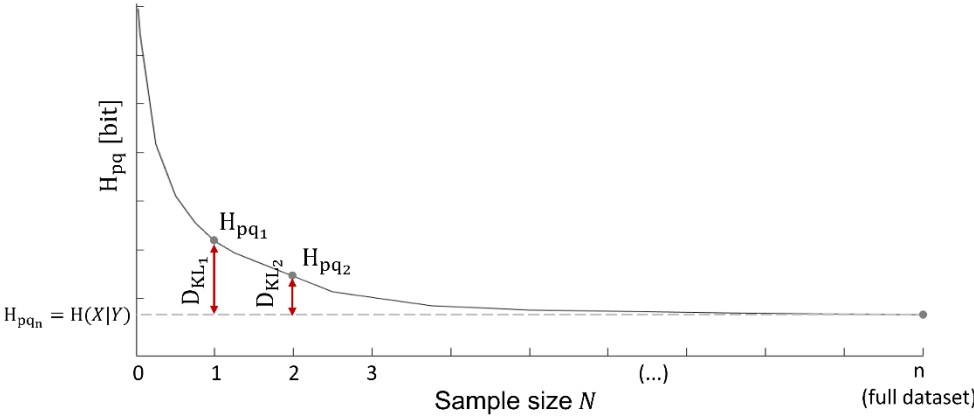

**Figure 2: Investigating the effect of sample size through Cross Entropy and Kullback-Leibler Divergence.**

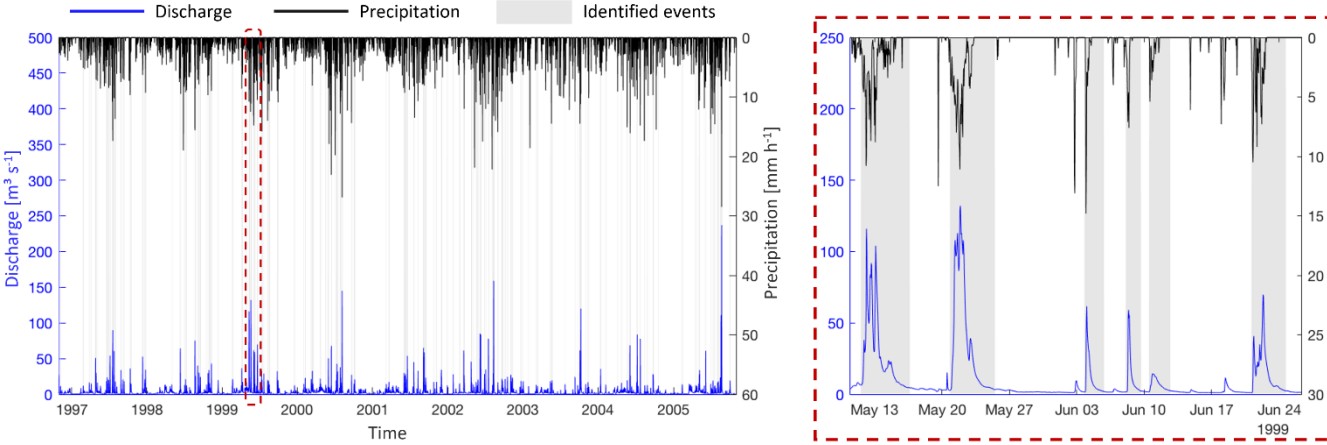

**Figure 3: Input data – Discharge, precipitation, user-based event classification. Overview of the time series on the left and detailed example on the right.**

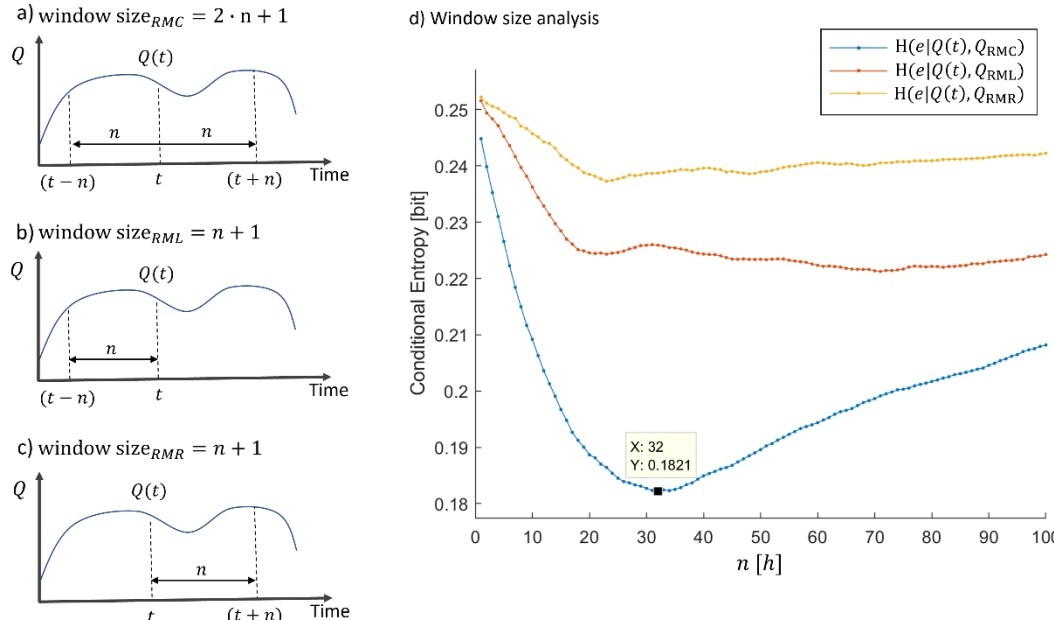

**Figure 4: Window size definitions for window types. a) $Q_{RMC}$, b) $Q_{RML}$, c) $Q_{RMR}$ window definitions and d) window size analysis.**

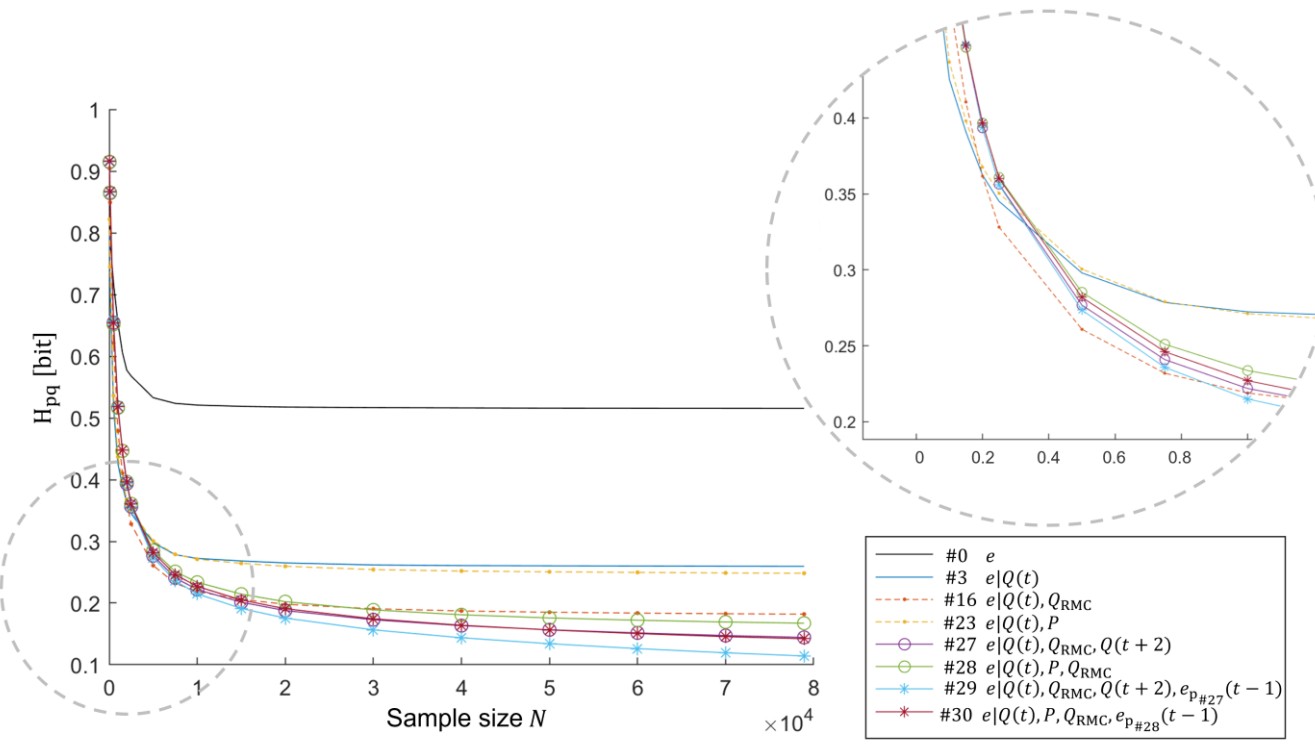

**Figure 5: Cross Entropy for models in Table 6 as a function of sample size.**

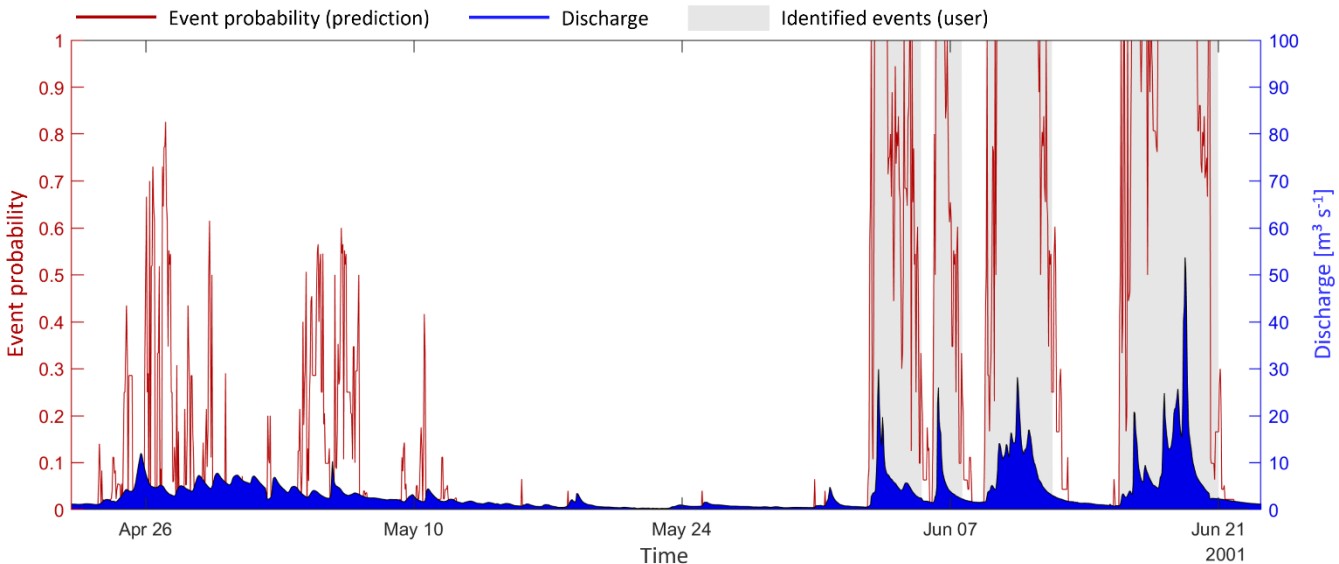

**Figure 6: Application I – Probabilistic prediction of 4-predictor model #29 (Table 5) to a subset of the training data.**

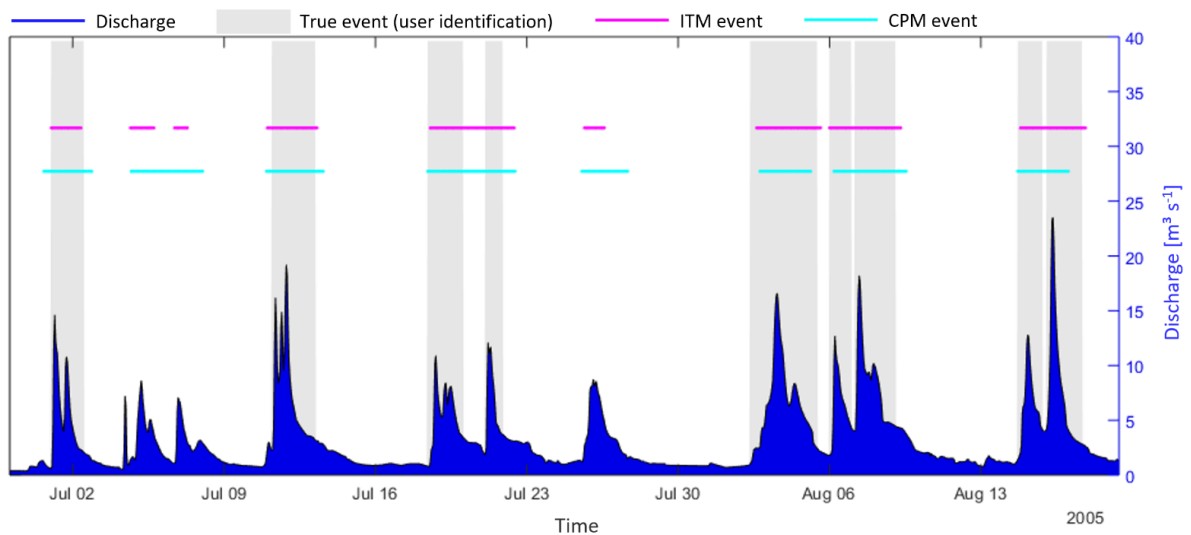

**Figure 7: Application II – Binary prediction of ITM and CPM to a subset of the testing set.**

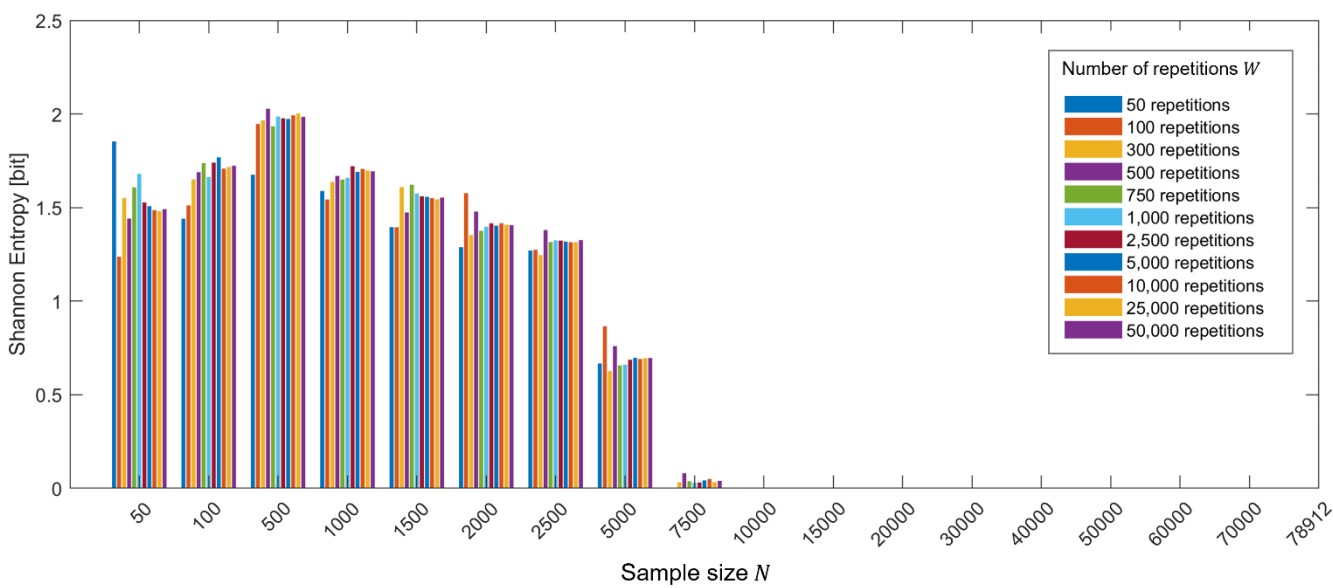

**Figure A1: Dispersion analysis of the Cross Entropy. The effect of the number of repetitions in the target model (#0 in Table 7).**