# Peer review of "Identifying rainfall-runoff events in discharge time series: A data-driven method based on information theory"

_Hydrology and Earth System Sciences, 2018_

## Referee Comment (RC1) · Y. Mei (Referee) · 13 Sep 2018

Summary

This manuscript describes a flood event identification method based on information theory. The idea of using data-based method in automatic event identification is novel; and few studies investigated event identification method for hourly flow time series. I am overall supportive of this research but I have some major concerns that need to be addressed by the authors. At this stage, I would assign major revision to the manuscript.

[Figure]

Comment

1. Precipitation information

Results of the manuscript show that precipitation at the current time step is not an informative predictor for flood event. I think this makes sense because there is always a lag time for the excess rainfall to travel to the outlet of the catchment. This has also been pointed out by Mei and Anagnostou (2015) with an event time lag parameter. Therefore, could a cumulative precipitation quantity that is produced based on a window with some size before the current time step help?

2. Size of evaluation window

Fig. 4: clearly, the window sizes are different in the center scheme to the forward and backward one and I observed different patterns of the conditional entropy with number of time step involved. I wonder why the authors did not use 2*n in the forward and backward scheme to make the window size consistent?

3. Probability threshold of event

Clearly, the selection of an optimal probability threshold is essential of the proposed method but the authors did not introduce a method of doing so. The only description is on P14 L10-15 and given the relatively short demonstration period, the authors arbitrarily select 75%. I wonder could such a threshold change with a) increasing number of time step and b) different basins?

4. Automation of the method

It can be seen from Figure 7 that there exist time steps associated with probability lower than 75% within the manually-identified events. This means that event identified by a user-defined fixed probability threshold are different than the manually identified one (i.e. the automatic one will have more events due to the existence of more separation time steps). So, do the authors have a method to skip those very-short discontinuous time steps so as to form longer events in an automatic manner?

**5. Detection of snowmelt event**

The authors show that the algorithm fell to detect the snowmelt event. I wonder is the inclusion of additional predictor can potentially help to extract the snowmelt event? For example, by adding in the in-situ or remote sensing-based observation of snow depth/extend, ground temperature and date of year, could the snowmelt event be identified?

**6. Comparison with existing method**

I think a comparison with the existing event identification methods could help to reveal the values of this newly-developed data-based method. If the authors would like to compare with Mei and Anagnostou (2015) method they may find the matlab codes on our GitHub profile (https://github.com/YiwenMei/Hydro_Seper). Another way to verify the method is to demonstrate the patterns of flood event parameters (e.g., runoff coefficient, time lag, baseflow) of events identified by the method.

**7. Potential usage of the method**

I did not see descriptions on potential usage of this method. Can this method be used to construct flood event database like for example the Shen et al. (2017, Comprehensive Database of Flood Events in the Contiguous United States from 2002 to 2013) constructed by the Characteristic Points Method introduced in Mei and Anagnostou (2015)?

---

## Author Comment (AC1) · 1 Oct 2018

We thank Dr. Yiwen Mei for reviewing our article and providing his feedbacks. Following, we have addressed all of his comments and discussed them. The observations were very helpful to identify some unclear issues regarding the method application.

Summary: This manuscript describes a flood event identification method based on information theory. The idea of using data-based method in automatic event identification is novel; and few studies investigated event identification method for hourly flow time series. I am overall supportive of this research but I have some major concerns that need to be addressed by the authors. At this stage, I would assign major revision to

the manuscript.

1. Precipitation information

Comment 1: Results of the manuscript show that precipitation at the current time step is not an informative predictor for flood event. I think this makes sense because there is always a lag time for the excess rainfall to travel to the outlet of the catchment. This has also been pointed out by Mei and Anagnostou (2015) with an event time lag parameter. Therefore, could a cumulative precipitation quantity that is produced based on a window with some size before the current time step help?

Response 1: We agree that aggregated precipitation is a potentially very useful predictor for event detection and will likely improve the results. The choice of predictors is an interactive and incremental process, and besides aggregated precipitation there may be other predictors which may eventually improve the results obtained so far. Thus, since the main point of the paper is to introduce the method without necessarily finding the perfect predictive model, and since precipitation data are often not available for analysis, we consider that the presented application is sufficient to demonstrate the potential of the approach. We suggest adding to a revised version of the manuscript a short discussion on the potential of using aggregated precipitation as a predictor.

2. Size of evaluation window

Comment 2: Fig. 4: clearly, the window sizes are different in the center scheme to the forward and backward one and I observed different patterns of the conditional entropy with number of time step involved. I wonder why the authors did not use 2*n in the forward and backward scheme to make the window size consistent?

Response 2: We have designed this parameter in a way such as to explore all possible window sizes at the finest resolution, while avoiding problems such as centering with odd window sizes. The "n" for each predictor was selected individually, and only displayed simultaneously on the same graph. The suggestion by the referee can be met

by simply using window size as x-axis position instead of "n" in the graph. We suggest that neither of the two choices entails a particular advantage or disadvantage. Thus, considering that the results are clearly visible and that the results will remain the same, we prefer to keep it as it is.

3. Probability threshold of event

Comment 3: Clearly, the selection of an optimal probability threshold is essential of the proposed method but the authors did not introduce a method of doing so. The only description is on P14 L10-15 and given the relatively short demonstration period, the authors arbitrarily select 75%. I wonder could such a threshold change with a) increasing number of time step and b) different basins?

Response 3: Indeed the threshold could be optimized, and it might be different for different datasets. We chose 75% as a threshold rather ad hoc, and only to demonstrate that it is possible to convert by choosing threshold the probabilistic prediction to a binary one if so desired. If there is no particular reason to do so, it will always be better to keep the probabilistic result as the binary transformation includes loss of information. That said, we agree with the referee that if a binary result is desired, the choice of threshold can and should be found by optimization. If the editor finds this a useful addition, we will do the related analysis (optimization by maximizing the number of hits in a contingency table) and add it to the manuscript. This will add about $\frac{3}{4}$ of a page to the manuscript (contingency table plus explanations).

4. Automation of the method action

Comment 4: It can be seen from Figure 7 that there exist time steps associated with probability lower than 75% within the manually-identified events. This means that event identified by a user-defined fixed probability threshold are different than the manually identified one (i.e. the automatic one will have more events due to the existence of more separation time steps). So, do the authors have a method to skip those very-short discontinuous time steps so as to form longer events in an automatic manner?

Response 4: We agree that in general it is desirable to have non-interrupted events of realistic length, but would like to mention that for some use-cases it is not relevant (e.g. if only the number of time steps classified as event is of interest). However, we see the use-cases where this property is of interest. There are several ways to address this in our method: The first is to include a memory effect in the classification by applying a recursive predictor ep(t-1). Comparable to a Markov model, this helps the model to better 'stick' to a classification after a transition from event to no-event or vice versa. While we already present and discuss such a model in the manuscript, the memory effect could be further enhanced if required by adding more recursive predictors (t-2, t-3, etc.). An alternative option would be to increase event coherence in a postprocessing step with an autoregressive model, with model parameters found by maximizing agreement with the observed events. We prefer the first method as it simply adds more predictors instead of adding another model component. We suggest describing these two options in a revised version of the manuscript.

5. Detection of snowmelt event

Comment 5: The authors show that the algorithm fell to detect the snowmelt event. I wonder is the inclusion of additional predictor can potentially help to extract the snowmelt event? For example, by adding in the in-situ or remote sensing-based observation of snow depth/extend, ground temperature and date of year, could the snowmelt event be identified?

Response 5: One of the strengths of the data-based approach we describe is that it accepts any kind of additional predictors such as air temperature, nitrate concentrations, etc. We agree with the referee that snow depth (or depth change) could be a potentially very useful predictor to identify snowmelt events. As mentioned in comment #1, we suggest that doing so would add another facet of application to the manuscript, but would not add to the method description as such. We therefore suggest adding a brief discussion of this topic in a revised version of the manuscript, but not an application.

**6. Comparison with existing method**

Comment 6: I think a comparison with the existing event identification methods could help to reveal the values of this newly-developed data-based method. If the authors would like to compare with Mei and Anagnostou (2015) method they may find the matlab codes on our GitHub profile (https://github.com/YiwenMei/Hydro_Seper). Another way to verify the method is to demonstrate the patterns of flood event parameters (e.g., runoff coefficient, time lag, baseflow) of events identified by the method.

Response 6: We agree that a comparison with existing method adds a valuable additional perspective to the study. If the Editor agrees, we will apply the Mei and Anagnostou method and compare results to the binary transforms of our probabilistic predictions via contingency tables and time-series plots. This will add at least one page to the manuscript.

**7. Potential usage of the method**

Comment 7: I did not see descriptions on potential usage of this method. Can this method be used to construct flood event database like for example the Shen et al. (2017, Comprehensive Database of Flood Events in the Contiguous United States from 2002 to 2013) constructed by the Characteristic Points Method introduced in Mei and Anagnostou (2015)?

Response 7: Given sufficient data to learning, we believe it is possible to construct such a database with our method. For each gauge in the database an individually optimized set of predictors could be identified. In addition, since we are dealing with a data-driven approach and avoiding parametrizations such as equations or indexes, the more event categories we aspire to classify, the more data will be required. Potential usages of the method were mentioned at the beginning of the introduction (P2 L8-34). We suggest adding the usage suggested by the referee to it.

427, 2018.

---

## Referee Comment (RC2) · Anonymous Referee #2 · 27 Oct 2018

Review of hess-2018-427

Identifying rainfall-runoff events in discharge time series: A data-driven method based on Information Theory

Stephanie Thiesen, Paul Darscheid, Uwe Ehret

Summary

——-

This paper discusses data-driven modeling approaches to identifying rainfall-runoff events. The authors construct probabilistic models using combinations of precipita-

tion and discharge to classify whether or not a given discharge should be considered an "event". Multiple models, with varying number of predictors are constructed. Each of these models are ranked by the reduction in uncertainty of the user-classified data. Overall, the paper has novel approaches and contains interesting results. However, in its current form the paper lacks several clarifying details and analyses and I cannot recommend publication in its current form. I suggest major revisions before the manuscript may be acceptable for publication,

Major comments

––––––––

1. The paper goes into detail with regards to hypothesis selection, model construction, and model evaluation, but lacks necessary details on model analysis. Model evaluation uses reduction in uncertainty of the user-defined classification, making this a supervised learning approach. However, there is very little analysis of how well the resulting models perform on new data. Figure 7 shows the application of the 4-predictor model, but it is unclear whether the application is on data that the model was trained on. A clear application of the resulting model on data which was not used during the model selection process is necessary to build credibility of the technique.

2. Motivations in this paper claim that existing automated event detection techniques are not adequate. In order to understand the benefits of this new technique there should be a comparison with one or more alternatively generated classifications. As currently written, it is unclear what baseline is used to decide that the method is good and whether the method is better than existing methods.

3. Figures 2, 5, and 6 don't provide very much insight. All simply show the convergence of the sub-sampled data to the overall data as sample sizes increase. Further, in figure 6 if the bar groupings are normalized by $\frac{D_{kl}}{H(X|Y)}[N=50]$ all of the curves would fall on top of each other. This is just an illustration of the ratio of the number of data points to the number of bins the estimators use. The caption text is also unclear,

as "curse of dimensionality" is not a formal quantity.

4. Page 7 line 23 claims that the prediction is not biased, but it is known that histogram-based entropy estimators systematically overpredict entropy [1]. This issue is particularly large in more than 3 dimensions [2]. If there is a systematic bias in the entropy estimation, it seems likely that the underlying probability distributions are then biased systematically as well. This is particularly true of the 4-predictor model.

Minor comments

————————

* Neither information theory nor curse of dimensionality need to be capitalized throughout.

* p.1 l.23: In the abstract it is unclear what "relative magnitude of discharge in a 65-hour time window" means. "Relative" to what?

* p.2 l.3: The quote from Chow 1988 appears to be missing a word, "... physiographic and climatic [word missing] that govern..."

* p.2 l.5-7: Aren't i) and iv) basically the same?

* p.2 l.20-29: The discussion of the history of event detection doesn't provide a particularly historic view. Event-detection (and baseflow separation) have a history that goes back a lot further than 2006.

* p.3 l.11-15: Claims about the bias and confidence from data driven methods need citation.

* p.4 l.1: It is unclear what is meant by "1:1" mapping between target and predictor data means on page 4 line 1.

* p.4 l.21: What happens when the system is not stationary?

* Equation 2 has an errant dot before $\log_2$.

* Page 6 line 26: good -> well.

* Section 2.3.1: The phrase "over the same underlying set of events" is unclear.

* The last sentence of page 6 requires a citation as well as clarification of what is meant by "how hard the Curse of Dimensionality hits"

* There are multiple (consistent) definitions of cross entropy.

* p.7 l.24-25: How do you determine whether "appropriate binning choice were made"?

* p.8 l.2: the area of the catchment should be in $km^2$.

* p.8 l.7: The arguments of the KL divergence need to be explained.

* Throughout section 3 discharge units should be $m^3 s^{-1}$

* Section 3.1: How is snow taken into account in the discharge time series (since this changes the timing between precipitation and streamflow)?

* Section 3.2.5: One sentence sections are rather odd.

* In the application of the method to a new time series, what happens when you encounter conditions that did not previously occur and which are outside the range of your empirical PMF?

* The heading for section 4.2 is vague (non-descriptive)

* p.12 l.19: "Computationally expensive" - what does that mean in this context? 2 minutes on a laptop or a week on a 30 thousand core computing cluster?

* Conclusions: The first part of the conclusions is just a summary of the paper. I think this can be shortened.

References
————-

[1] Steuer, R., Kurths, J., Daub, C. O., Weiseand, J., & Selbig, J. (2002). The mutual information: Detecting and evaluating depencies between variables. Bioinformatics, 18(2), S231–S240.

[2] Hlaváčková-Schindler, K., Paluš, M., Vejmelka, M., & Bhattacharya, J. (2007). Causality detection based on information-theoretic approaches in time series analysis. Physics Reports, 441(1), 1–46. https://doi.org/10.1016/j.physrep.2006.12.004

---

## Author Comment (AC2) · 19 Nov 2018

We thank the second referee for cautiously reviewing our article and providing his/her feedback. Following, we have addressed all of his/her comments and discussed them. The observations were complementary to referee #1 and very helpful to identify some unclear issues.

Summary: This paper discusses data-driven modeling approaches to identifying rainfall-runoff events. The authors construct probabilistic models using combinations of precipitation and discharge to classify whether or not a given discharge should be considered an "event". Multiple models, with varying number of predictors are constructed.

[Figure]

Each of these models are ranked by the reduction in uncertainty of the user-classified data. Overall, the paper has novel approaches and contains interesting results. However, in its current form the paper lacks several clarifying details and analyses and I cannot recommend publication in its current form. I suggest major revisions before the manuscript may be acceptable for publication.

Major Comments:

Comment 1: The paper goes into detail with regards to hypothesis selection, model construction, and model evaluation, but lacks necessary details on model analysis. Model evaluation uses reduction in uncertainty of the user-defined classification, making this a supervised learning approach. However, there is very little analysis of how well the resulting models perform on new data. Figure 7 shows the application of the 4-predictor model, but it is unclear whether the application is on data that the model was trained on. A clear application of the resulting model on data which was not used during the model selection process is necessary to build credibility of the technique.

Response 1: The application presented in Figure 7 is indeed on a subset of the training data, as mentioned in the manuscript (page 14 line 6, and legend of Figure 7). From this and the following major comments, as well as comments made by referee #1 we conclude that there is a need to both better explain and justify the approach we took in the paper for training and evaluating (t+e in the following) the models, and to also include for comparison approaches to t+e the reader is more familiar with, and to compare in this standard setting our model to benchmark models. We start by explaining our approach, and will then propose additions to the manuscript. Our approach to t+e is summarized in Fig. 2, and corresponding results are shown in Figure 5. What the figures show is a summary of a large number of t+e operations, where from the available data randomly chosen subsets of various sizes are used for calibration/supervised learning, and the resulting model is then applied to and evaluated on all data. So the referee is correct in stating that the model is partly evaluated on data it has seen during learning, but at the same time it is also evaluated against data it has not seen yet. So

this is different from a standard split sample approach, where the data sets for training and evaluation are mutually exclusive. However, we would like to argue that the standard split sample approach has the problem of using different data sets of different sizes for validation (depending on the choice and length of the calibration period), which makes comparison of results difficult. Also, in our approach for a given length of training period we conduct many tests and average the results, which yields more robust results than the standard approach of dividing the available data only once into a calibration and validation period, and providing results only from that single test (however, we are aware that for split sampling also more elaborate approaches exist which include more than a single split). So we think that our approach to t+e has some advantages to offer, among them convergence to the best achievable result for increasing sample sizes, which we would like to keep, and we would like to stress that our approach also includes an evaluation of the model on data it has not seen during training, even though this portion varies with the chosen sample size. Nevertheless, we think it will be beneficial to include a standard approach to t+e into the paper: Firstly because it will be easier to explain the specialties of our approach by contrasting it to an established approach, secondly because it will be easier for the reader to judge the model quality by comparing it to own experiences and to the benchmark model in a familiar setting. So we suggest adding a new section to a revised version of the manuscript, in which we will train our model and, as a benchmark, the method proposed by Mei and Anagnostou (2015), in a single training period (the first 9 years of the available data) and then apply, evaluate and compare the models in the remaining last year.

Comment 2: Motivations in this paper claim that existing automated event detection techniques are not adequate. In order to understand the benefits of this new technique there should be a comparison with one or more alternatively generated classifications. As currently written, it is unclear what baseline is used to decide that the method is good and whether the method is better than existing methods.

Response 2: We agree and suggest applying as a benchmark the method proposed
by Mei and Anagnostou (2015) and discuss the relative benefits of each method via comparison. Please also see our reply to comment 1.

Comment 3: Figures 2, 5, and 6 don't provide very much insight. All simply show the convergence of the sub-sampled data to the overall data as sample sizes increase. Further, in figure 6 if the bar groupings are normalized by $\frac{D_{kl}}{H(X|Y)}[N=50]$ all of the curves would fall on top of each other. This is just an illustration of the ratio of the number of data points to the number of bins the estimators use. The caption text is also unclear, as "curse of dimensionality" is not a formal quantity.

Response 3: Please see our response to comment 1: We would like to maintain that Fig. 2 and 5 provide essential summaries both of our approach to evaluate models (Fig. 2) and actual results for the given data (Fig. 5). Both figures contain a joint visualization of model analysis and model evaluation, and at the same time provide the opportunity of comparing models with different numbers of predictors. That is, they provide an opportunity to decide, for a given amount of data, which number of predictors is optimal in the sense of avoiding both ignoring available information (by choosing too few predictors) and overfitting (by choosing too many predictors). Fig. 6 was meant as an illustration of our concept to formalize (and make objective) the decision what amount of learning data provides a sufficiently good model for the entire data set. The percentage Dkl/H(X|Y) measures what share of total predictive uncertainty is due to a less-than-optimal model (because it was trained only on a subset of the data). If a user decides on such a limit (e.g. 5%), the minimum amount of training data to assure this can be read from the figure for models with different numbers of predictors. However, we agree that in its current form Fig. 6 may be overly detailed. We suggest removing the figure, and instead add to table 7, column 'Sample size where Dkl/H(X|Y)$\leq$5% the relative sample size in addition to the absolute sample size.

Comment 4: Page 7 line 23 claims that the prediction is not biased, but it is known that histogram based entropy estimators systematically overpredict entropy [1]. This issue is particularly large in more than 3 dimensions [2]. If there is a systematic bias in the
entropy estimation, it seems likely that the underlying probability distributions are then biased systematically as well. This is particularly true of the 4-predictor model.

Response 4: The referee raises several interesting and interconnected aspects of calculating information measures from limited samples and/or binned representations of distributions. The effect of binning: In general, the choice of binning affects the absolute values of information measures of a single variate (entropy) as well as information measures of dependency between variates (conditional entropy, mutual information). If this effect is not of interest, it can be avoided by sticking to the same binning throughout an analysis, such that only the relative, not the absolute magnitude of the information measures matter. This is what we did in our study, as we were not interested in the 'true' information measures calculated from 'true' continuous data (or at least data mapped to very high-resolution bins), but on the relative magnitudes of the information measures for various choices and numbers of predictors. The effect of limited samples: Computing information measures from limited samples can indeed introduce biases in the estimation of information measures: entropy is systematically underestimated albeit the underestimation is bounded (see Paninski, 2003), mutual information is systematically overestimated (see Steuer et al., 2002 as mentioned by the referee). In the extreme, if the sample consists of a single observation or single pair of observations, estimated entropy will be zero, and there will be full mutual information (knowing one of the paired values unambiguously identifies the other). So especially in the case of small training data sets, if we would assume that the information measures derived in the training data set would also hold for the case of applying the model to new data, we would systematically over- or underestimate the true values. However, in our study we evaluate the performance of each model always against the same reference, which is the full data set. The measure we use (cross entropy) incorporates both the agreement of true relation among the data and the relation as expressed by the model via Kullback-Leibler divergence and the predictability of the target given the predictors for the full data set (conditional entropy). As a consequence, the negative effects of learning from limited samples is considered in the relative ranking of the various tested mod-

els, which is what we want. Altogether, we suggest that including the above discussion in the manuscript would be beyond its scope. So in order to avoid misinterpretations, we suggest rephrasing the text in section 2.4 in a revised version of the manuscript: "[. . .]. The model returns a probabilistic representation of the target value. If the model was trained on all available data, and is applied within the domain of these data, the predictions will be unbiased and neither over- nor underconfident. If instead a model using deterministic functions is trained and applied in the same manner, the resulting single-valued predictions may also be unbiased, but due to their single-value nature will surely be overconfident.".

Minor Comments:

Comment 5: Neither information theory nor curse of dimensionality need to be capitalized throughout.

Response 5: Thanks. We will adapt the writing style in a revised version of the manuscript.

Comment 6: p.1 l.23: In the abstract it is unclear what "relative magnitude of discharge in a 65-hour time window" means. "Relative" to what?

Response 6: We mean "relative to all values in the time window". We suggest rephrasing the sentence "[. . .]: discharge from two distinct time steps, the relative magnitude of discharge compared to all discharge values in a surrounding 65-hour time window, and event predictions from the previous time step.

Comment 7: p.2 l.3: The quote from Chow 1988 appears to be missing a word, "... physiographic and climatic [word missing] that govern..."

Response 7: Thank you. The correct quote is indeed "[...] physiographic and climatic characteristics that govern [...]". We will include the correct quote in a revised version of the manuscript.

Comment 8: p.2 l.5-7: Aren't i) and iv) basically the same?

[Figure]

Response 8: We rethought the arguments made here and suggest rephrasing the sentence: "Discharge time series are a fundamental component of hydrological learning and prediction since they i) are relatively easy-to-obtain, in high quality and from widespread and long-existing observation networks. ii) carry robust and integral information about the catchment state, and iii) are an important target quantity for hydrological prediction and decision-making."

Comment 9: p.2 l.20-29: The discussion of the history of event detection doesn't provide a particularly historic view. Event-detection (and baseflow separation) have a history that goes back a lot further than 2006.

Response 9: Correct. In our discussion we focused on relatively recent techniques suitable for automated event detection. In order to put the paper in the proper historical context, we suggest adding in a revised version of the manuscript a brief overview on older methods of event detection and baseflow separation.

Comment 10: p.3 l.11-15: Claims about the bias and confidence from data driven methods need citation.

Response 10: We suggest rephrasing to clarify what we wanted to say: "Predictions based on probabilistic models that learn relations among data directly from the data, with few or no prior assumptions about the nature of these relations, are less bias-prone (because there are no prior assumptions potentially obstructing convergence towards observed mean behavior), and less likely to be overconfident compared to established models (because applying deterministic models is still standard hydrological practice, and they are overconfident in all but the very few cases of perfect models). This applies at least if there is sufficient data to learn from, appropriate binning choices were made and the application remains within the domain of the data that were used for learning." We suggest that these claims are self-evident and do not require a citation.

Comment 11: p.4 l.1: It is unclear what is meant by "1:1" mapping between target and predictor data means on page 4 line 1.

Response 11: It means that each target has exactly one corresponding predictor, i.e., one particular value of target is related to one particular value of predictor (in contrast to 1:n or n:m relationships). We suggest including a more elaborate explanation in a revised version of the manuscript.

Comment 12: p.4 l.21: What happens when the system is not stationary?

Response 12: If the system is non-stationary, i.e. system properties change with time, the inconsistency between the learning and the prediction situation will result in additional predictive uncertainty. The problems associated with predictions of nonstationary systems apply to all modeling approaches. If a stable trend can be identified, a possible countermeasure is to do learning and prediction on detrended data and then re-impose the trend in a post-processing step. We suggest adding this sentence to a revised version of the manuscript.

Comment 13: Equation 2 has an errant dot before $\log_2$.

Response 13: Thank you. We will correct this in a revised version of the manuscript.

Comment 14: Page 6 line 26: good -> well.

Response 14: Thank you. We will correct this in a revised version of the manuscript.

Comment 15: Section 2.3.1: The phrase "over the same underlying set of events" is unclear.

Response 15: We agree that the meaning is not clear. We suggest rephrasing "It is also possible to compare two probability distributions p and q."

Comment 16: The last sentence of page 6 requires a citation as well as clarification of what is meant by "how hard the Curse of Dimensionality hits"

Response 16: We have explained the Curse of Dimensionality in detail and with citations in the same section (p. 6 l.12-23), so we think there is no need to further explain it at the end of page 6. However, to make clearer what we mean we suggest rephrasing
the sentence to "[...] i.e., it is a measure of the impact of the Curse of Dimensionality.".

Comment 17: There are multiple (consistent) definitions of cross entropy.

Response 17: Sorry, we do not understand this comment. In the text we provide several definitions and interpretations of cross entropy to provide the reader with a comprehensive perspective of the subject.

Comment 18: p.7 l.24-25: How do you determine whether "appropriate binning choice were made"?

Response 18: A befitting binning choice is indeed a subject of ongoing research, see e.g. Gong et al. (2014) or Pechlivanidis et al. (2016). In general terms, as mentioned in the manuscript (page 4, lines 8-16) an appropriate binning choice yields bins which are neither too narrow (which leads to a overfitted model) nor too wide (which leads to overly smoothed histograms, which will introduce a significant amount of bias and also discards information about the high resolution details of the distribution). We suggest rephrasing the sentence in a revised version of the manuscript: "[...], and appropriate binning choices were made (see the related discussion in section 2.2).".

Comment 19: p.8 l.2: the area of the catchment should be in $km^2$.

Response 19: Sorry, we do not understand this comment. The catchment area is already expressed in km^2.

Comment 20: p.8 l.7: The arguments of the KL divergence need to be explained.

Response 20: Sorry, we do not understand this comment. We could not find arguments regarding KL divergence in the mentioned page.

Comment 21: Throughout section 3 discharge units should be $m^3 s^{-1}$

Response 21: Sorry, we do not understand this comment. Discharge values are already expressed in m^3 s^-1.

[Figure]

Comment 22: Section 3.1: How is snow taken into account in the discharge time series (since this changes the timing between precipitation and streamflow)?

Response 22: Please note that referee #1 had a similar question, so we mainly provide the same reply here as to referee #1, comment 5. Effects of snow accumulation and melting on discharge events were not the central point of the study. Thus, they were not explicitly considered or classified by the model. One of the strengths of the data-based approach we describe is that it potentially accepts any kind of additional predictors such as air temperature, nitrate concentrations, or snow. We agree with the referee that snow-related observations could be a potentially very useful predictor to identify snowmelt events. However, we suggest that doing so would add another facet of application to the manuscript, but would not add to the method description as such, which is the main goal of our paper. We therefore suggest adding a brief discussion of this topic in a revised version of the manuscript, but not an application.

Comment 23: Section 3.2.5: One sentence sections are rather odd.

Response 23: We agree that this looks odd. To avoid it we considered merging subsections of section 3.2. However, due to the nature of the different predictors (many discharge-based predictors and few others), in order to merge precipitation we'd have to merge them all, which would greatly reduce readability. We think that the one-sentence section is the lesser evil and suggest keeping the subsection structure in 3.2 as it is.

Comment 24: In the application of the method to a new time series, what happens when you encounter conditions that did not previously occur and which are outside the range of your empirical PMF?

Response 24: Good point. If the conditions are outside of the range of the empirical PMF, the model will fail, i.e. it will provide no prediction. The same problem will also occur if the conditions are within the range of the empirical PMF, but have never been observed, at least not as seen through the filter of the chosen binning. In that case, the
predictive distribution of the target (event Yes/No) will also be empty. If 'no answer' by the model is not acceptable, several methods exist to guarantee an answer, however at the cost of reduced precision:

* Coarse graining: The PMF can be rebuild with fewer, wider bins and an extension of the range (e.g. by merging neighboring bins) until the model provides an answer to the predictive setting. This way the PMF is still purely based on the observed data, but the resulting predictive distributions will be more spread-out as they integrate more observations from a larger range of observed situations. Similar methods to avoid empty bins by adjusting the binning have been proposed for example by Darbellay and Vajda (1999), Knuth (2013) and Pechlivanidis et al. (2016).

* Gap-filling: A widely applied alternative is to maintain the binning and fill the empty bins with non-zero values based on a deemed-to-be-reasonable assumption on the shape of the PMF if more data were available. Approaches comprise adding one counter to each zero-probability bin of the sample histogram, adding a small probability to the sample PDF, smoothing methods such as Kernel-density smoothing (Blower et al. , 2002; Simonoff, 1996), or Bayesian approaches based on the Dirichlet and Multinomial distribution, or a Maximum-Entropy Method recently suggested by Darscheid et al. (2018). All of these methods are applicable both for the extrapolation case mentioned by the referee and the interpolation case. We suggest adding to a revised version of the manuscript a short discussion of this topic as we think it is of interest for the reader.

Comment 25: The heading for section 4.2 is vague (non-descriptive)

Response 25: We chose this section header in relation to that of section 4.1 (model performance for the full data set) to emphasize the difference between the two. In order to maintain this emphasis, we prefer to keep the header as it is.

Comment 26: p.12 l.19: "Computationally expensive" - what does that mean in this context? 2 minutes on a laptop or a week on a 30 thousand core computing cluster?

Response 26: In our work it means one standard PC running for around 300 hours to analyze a 5- dimensional model (18 different sample sizes, 500 repetitions of each sample size). As it is well known that resampling-based analysis techniques tend to be computationally expensive, we suggest keeping the statement in the text as general as it is.

Comment 27: Conclusions: The first part of the conclusions is just a summary of the paper. I think this can be shortened.

Response 27: We agree that the summary can be shortened and suggest to do so in a revised version of the manuscript. However, we think it is important to give the reader a short wrap-up of what was done before the conclusions are drawn, so we will not abandon the summary altogether. Also, in order to better reflect the content of the section, we suggest changing the section header from 'Conclusions' to 'Summary and conclusions'.

References:

Blower, G.; Kelsall, J.E. Nonlinear Kernel Density Estimation for Binned Data: Convergence in Entropy. Bernoulli 2002, 8, 423–449.

Darbellay, G.A., Vajda, I. Estimation of the information by an adaptive partitioning of the observation space. IEEE Trans. Inf. Theory 1999, 45, 1315–1321.

Darscheid, P., Guthke, A., & Ehret, U. (2018). A Maximum-Entropy Method to Estimate Discrete Distributions from Samples Ensuring Nonzero Probabilities. Entropy, 20, 601. https://doi.org/10.3390/e20080601

[2] Hlaváčková-Schindler, K., Paluš, M., Vejmelka, M., & Bhattacharya, J. (2007). Causality detection based on information-theoretic approaches in time series analysis. Physics Reports, 441(1), 1–46. https://doi.org/10.1016/j.physrep.2006.12.004

Knuth, K.H. Optimal Data-Based Binning for Histograms. arXiv 2013, arXiv:physics/0605197v2.

Mei, Y., Anagnostou, E. N. (2015), A hydrograph separation method based on information from rainfall and runoff records, Journal of Hydrology, 523, 636-649.

Paninski, L. (2003): Estimation of entropy and mutual information. Neural Comput., 15(6): 1191-1253.

Pechlivanidis, I.G.; Jackson, B.; McMillan, H.; Gupta, H.V. Robust informational entropy-based descriptors of flow in catchment hydrology. Hydrol. Sci. J. 2016, 61, 1–18.

Simonoff, J.S. Smoothing Methods in Statistics; Springer: Berlin/Heidelberg, Germany, 1996.

[1] Steuer, R., Kurths, J., Daub, C. O., Weiseand, J., & Selbig, J. (2002). The mutual information: Detecting and evaluating dependencies between variables. Bioinformatics, 18(2), S231–S240.

---

## Editor Comment (EC1) · B. Schaefli (Editor) · 22 Nov 2018

Both reviewers state that the presented method for flood event identification based on information theory is interesting and contains novel aspects; they also both discuss that the manuscript requires methodological clarification at several instances and a comparison with another method (as benchmark). Accordingly, I invite the authors to revise their manuscript along their responses in the public discussion. The revised version will go into re-review.

---

## Author Response (AR1)

**Dear Mrs. Schaefli,**

**The current document consolidates the referee comments, the author responses posted so far and the actions taken in the major revision of the manuscript. It is organized according to the following legend:**

- Referee comments
- Authors' responses from the previous stage
- Authors' major revisions and responses for the new version of the manuscript

**After this section of comments and responses, you can find the manuscript pdf with the tracked revision. Please note that the page numbers in our responses refer to the revised, track-free version of the manuscript.**

**Best regards,**
**Stephanie Thiesen, Uwe Ehret and Paul Darscheid**

**Referee #1 | Dr. Yiwen Mei**

**Comment 1:** Results of the manuscript show that precipitation at the current time step is not an informative predictor for flood event. I think this makes sense because there is always a lag time for the excess rainfall to travel to the outlet of the catchment. This has also been pointed out by Mei and Anagnostou (2015) with an event time lag parameter. Therefore, could a cumulative precipitation quantity that is produced based on a window with some size before the current time step help?

**Response 1**: We agree that aggregated precipitation is a potentially very useful predictor for event detection and will likely improve the results. The choice of predictors is an interactive and incremental process, and besides aggregated precipitation there may be other predictors which may eventually improve the results obtained so far. Thus, since the main point of the paper is to introduce the method without necessarily finding the perfect predictive model, and since precipitation data are often not available for analysis, we consider that the presented application is sufficient to demonstrate the potential of the approach. We added to the new version of the manuscript a brief discussion on the potential of using aggregated precipitation as a predictor and snowmelt events (Section 4.1.3, p. 17, l. 4-8).

**Comment 2**: Fig. 4: clearly, the window sizes are different in the center scheme to the forward and backward one and I observed different patterns of the conditional entropy with number of time step involved.  I wonder why the authors did not use 2*n in the forward and backward scheme to make the window size consistent?

**Response 2**: We have designed this parameter in a way such as to explore all possible window sizes at the finest resolution, while avoiding problems such as centering with odd window sizes.

The "n" for each predictor was selected individually, and only displayed simultaneously on the same graph. The suggestion by the referee can be met by simply using window size as x-axis position instead of "n" in the graph. We suggest that neither of the two choices entails a particular advantage or disadvantage. Thus, considering that the results are clearly visible and that the results will remain the same, there were no changes in the new version of the manuscript.

**Comment 3**: Clearly, the selection of an optimal probability threshold is essential of the proposed method but the authors did not introduce a method of doing so. The only description is on P14 L10-15 and given the relatively short demonstration period, the authors arbitrarily select 75%. I wonder could such a threshold change with a) increasing number of time step and b) different basins?

**Response 3**: Indeed the threshold could be optimized, and it might be different for different datasets. We chose 75% as a threshold rather ad hoc, and only to demonstrate that it is possible to convert by choosing threshold the probabilistic prediction to a binary one if so desired. If there is no particular reason to do so, it will always be better to keep the probabilistic result as the binary transformation includes loss of information. That said, we agree with the referee that if a binary result is desired, the choice of threshold can and should be found by optimization. The binarization procedure was added as part of the benchmarking comparison between the proposed ITM (information theory method) and the CPM (characteristic point method, proposed by Mei and Anagnostou, 2015) in Section 3.3, and its application can be observed in Section 4.2.

**Comment 4**: It can be seen from Figure 7 that there exist time steps associated with probability lower than 75% within the manually-identified events. This means that event identified by a user-defined fixed probability threshold are different than the manually identified one (i.e. the automatic one will have more events due to the existence of more separation time steps). So, do the authors have a method to skip those very-short discontinuous time steps so as to form longer events in an automatic manner?

**Response 4**: We agree that in general it is desirable to have non-interrupted events of realistic length, but would like to mention that for some use-cases it is not relevant (e.g. if only the number of time steps classified as event is of interest).
However, we see the use-cases where this property is of interest. There are several ways to address this in our method: The first is to include a memory effect in the classification by applying a recursive predictor $e_p(t-1)$. Comparable to a Markov model, this helps the model to better 'stick' to a classification after a transition from event to no-event or vice versa. While we already present and discuss such a model in the manuscript, the memory effect could be further enhanced if required by adding more recursive predictors (t-2, t-3, etc.). An alternative option would be to increase event coherence in a post-processing step with an autoregressive model, with model parameters found by maximizing agreement with the observed events. We prefer the first method as it simply adds more predictors instead of adding another model component. A brief discussion of the alternatives was incorporated at the end of Section 4.1.3 (p. 17, l. 9-15), but not an application.

**Comment 5**: The authors show that the algorithm fell to detect the snowmelt event. I wonder is the inclusion of additional predictor can potentially help to extract the snowmelt event? For example, by adding in the in-situ or remote sensing-based observation of snow depth/extend, ground temperature and date of year, could the snowmelt event be identified?

**Response 5**: One of the strengths of the data-based approach we describe is that it accepts any kind of additional predictors such as air temperature, nitrate concentrations, etc. We agree with the referee that snow depth (or depth change) could be a potentially very useful predictor to identify snowmelt events. As mentioned in comment #1, we suggest that doing so would add another facet of application to the manuscript, but would not add to the method description as such. Same as response #1. We added to the new version of the manuscript a brief discussion on the potential of using aggregated precipitation as a predictor and snowmelt events (Section 4.1.3, p. 17, l. 4-8).

**Comment 6**: I think a comparison with the existing event identification methods could help to reveal the values of this newly-developed data-based method. If the authors would like to compare with Mei and Anagnostou (2015) method they may find the matlab codes on our GitHub profile (https://github.com/YiwenMei/Hydro_Seper). Another way to verify the method is to demonstrate the patterns of flood event parameters (e.g., runoff coefficient, time lag, baseflow) of events identified by the method.

**Response 6**: We agree that a comparison with existing method adds a valuable additional perspective to the study. Thus, two new sections were added to describe (Section 3.3) and apply (4.2) the binary transformation of the probabilistic prediction, holdout cross-validation and comparative analysis between ITM and CPM.

**Comment 7**: I did not see descriptions on potential usage of this method. Can this method be used to construct flood event database like for example the Shen et al. (2017, Comprehensive Database of Flood Events in the Contiguous United States from 2002 to 2013) constructed by the Characteristic Points Method introduced in Mei and Anagnostou (2015)?

**Response 7**: Given sufficient data to learning, we believe it is possible to construct such a database with our method. For each gauge in the database an individually optimized set of predictors could be identified. In addition, since we are dealing with a data-driven approach and avoiding parametrizations such as equations or indexes, the more event categories we aspire to classify, the more data will be required. The discussion of the model potential usage was updated in Sections 1 (p. 3, l. 27-30) and Section 5 (p. 20, l. 11-22).

**Referee #2 | Anonymous**

**Comment 1**: The paper goes into detail with regards to hypothesis selection, model construction, and model evaluation, but lacks necessary details on model analysis. Model evaluation uses reduction in uncertainty of the user-defined classification, making this a supervised learning approach. However, there is very little analysis of how well the resulting

models perform on new data. Figure 7 shows the application of the 4-predictor model, but it is unclear whether the application is on data that the model was trained on. A clear application of the resulting model on data which was not used during the model selection process is necessary to build credibility of the technique.

**Response 1**: The application presented in Figure 7 is indeed on a subset of the training data, as mentioned in the manuscript (page 14 line 6, and legend of Figure 7).

From this and the following major comments, as well as comments made by referee #1 we conclude that there is a need to both better explain and justify the approach we took in the paper for training and evaluating (t+e in the following) the models, and to also include for comparison approaches to t+e the reader is more familiar with, and to compare in this standard setting our model to benchmark models.

We start by explaining our approach, and will then propose additions to the manuscript.

Our approach to t+e is summarized in Fig. 2, and corresponding results are shown in Figure 5. What the figures show is a summary of a large number of t+e operations, where from the available data randomly chosen subsets of various sizes are used for calibration/supervised learning, and the resulting model is then applied to and evaluated on all data. So the referee is correct in stating that the model is partly evaluated on data it has seen during learning, but at the same time it is also evaluated against data it has not seen yet. So this is different from a standard split sample approach, where the data sets for training and evaluation are mutually exclusive. However, we would like to argue that the standard split sample approach has the problem of using different data sets of different sizes for validation (depending on the choice and length of the calibration period), which makes comparison of results difficult. Also, in our approach for a given length of training period we conduct many tests and average the results, which yields more robust results than the standard approach of dividing the available data only once into a calibration and validation period, and providing results only from that single test (however, we are aware that for split sampling also more elaborate approaches exist which include more than a single split). So we think that our approach to t+e has some advantages to offer, among them convergence to the best achievable result for increasing sample sizes, which we would like to keep, and we would like to stress that our approach also includes an evaluation of the model on data it has not seen during training, even though this portion varies with the chosen sample size.

Nevertheless, we think it will be beneficial to include a standard approach to t+e into the paper: Firstly because it will be easier to explain the specialties of our approach by contrasting it to an established approach, secondly because it will be easier for the reader to judge the model quality by comparing it to own experiences and to the benchmark model in a familiar setting.

Thus, as mentioned in the response #1 of referee #1, two new sections were added to describe (Section 3.3) and apply (4.2) the binary transformation of the probabilistic prediction, holdout cross-validation and comparative analysis between ITM and CPM. Furthermore, an more detailed explanation regarding Fig. 2 was added in Section 2.3.2 (p. 7, l. 18-27).

**Comment 2**: Motivations in this paper claim that existing automated event detection techniques are not adequate. In order to understand the benefits of this new technique there should be a comparison with one or more alternatively generated classifications. As currently written, it is unclear what baseline is used to decide that the method is good

and whether the method is better than existing methods.

**Response 2:** We agree and suggest applying as a benchmark the method proposed by Mei and Anagnostou (2015) and discuss the relative benefits of each method via comparison. As mentioned in the previous answer, two new sections were added to describe (Section 3.3) and apply (4.2) the binary transformation of the probabilistic prediction, holdout cross-validation and comparative analysis between ITM and CPM.

**Comment 3**: Figures 2, 5, and 6 don't provide very much insight. All simply show the convergence of the sub-sampled data to the overall data as sample sizes increase. Further, in figure 6 if the bar groupings are normalized by $\frac{D_{kl}}{H(X|Y)}[N=50]$ all of the curves would fall on top of each other. This is just an illustration of the ratio of the number of data points to the number of bins the estimators use. The caption text is also unclear, as "curse of dimensionality" is not a formal quantity.

**Response 3**: Please see our response to comment 1: We would like to maintain that Fig. 2 and 5 provide essential summaries both of our approach to evaluate models (Fig. 2) and actual results for the given data (Fig. 5). Both figures contain a joint visualization of model analysis and model evaluation, and at the same time provide the opportunity of comparing models with different numbers of predictors. That is, they provide an opportunity to decide, for a given amount of data, which number of predictors is optimal in the sense of avoiding both ignoring available information (by choosing too few predictors) and overfitting (by choosing too many predictors). Fig. 2 and Fig. 5 were maintained and a more detailed explanation regarding Fig. 2 was added in Section 2.3.2 (p. 7, l. 18-27) and regarding Fig. 5 in Section 2.3.2 (p. 15, l. 16-18). Fig. 6 was meant as an illustration of our concept to formalize (and make objective) the decision what amount of learning data provides a sufficiently good model for the entire data set. The percentage Dkl/H(X|Y) measures what share of total predictive uncertainty is due to a less-than-optimal model (because it was trained only on a subset of the data). If a user decides on such a limit (e.g. 5%), the minimum amount of training data to assure this can be read from the figure for models with different numbers of predictors. However, we agree that in its current form Fig. 6 may be overly detailed. In the new version of the manuscript, we removed the Fig. 6, and instead added to Table 7 (Column 5) the relative sample size given the full data set, in addition to the absolute sample size.

**Comment 4**: Page 7 line 23 claims that the prediction is not biased, but it is known that histogram based entropy estimators systematically overpredict entropy [1]. This issue is particularly large in more than 3 dimensions [2]. If there is a systematic bias in the entropy estimation, it seems likely that the underlying probability distributions are then biased systematically as well. This is particularly true of the 4-predictor model.

**Response 4**: The referee raises several interesting and interconnected aspects of calculating information measures from limited samples and/or binned representations of distributions. The effect of binning: In general, the choice of binning affects the absolute values of information measures of a single variate (entropy) as well as information measures of dependency between variates (conditional entropy, mutual information). If this effect is not of interest, it can be avoided by sticking to the same binning throughout an analysis, such that only the relative, not the absolute magnitude of the information measures matter. This is what we did in our study, as

we were not interested in the 'true' information measures calculated from 'true' continuous data (or at least data mapped to very high-resolution bins), but on the relative magnitudes of the information measures for various choices and numbers of predictors.

The effect of limited samples: Computing information measures from limited samples can indeed introduce biases in the estimation of information measures: entropy is systematically underestimated albeit the underestimation is bounded (see Paninski, 2003), mutual information is systematically overestimated (see Steuer et al., 2002 as mentioned by the referee). In the extreme, if the sample consists of a single observation or single pair of observations, estimated entropy will be zero, and there will be full mutual information (knowing one of the paired values unambiguously identifies the other). So especially in the case of small training data sets, if we would assume that the information measures derived in the training data set would also hold for the case of applying the model to new data, we would systematically over- or underestimate the true values.

However, in our study we evaluate the performance of each model always against the same reference, which is the full data set. The measure we use (cross entropy) incorporates both the agreement of true relation among the data and the relation as expressed by the model via Kullback-Leibler divergence and the predictability of the target given the predictors for the full data set (conditional entropy). As a consequence, the negative effects of learning from limited samples is considered in the relative ranking of the various tested models, which is what we want.

Altogether, we suggest that including the above discussion in the manuscript would be beyond its scope. So in order to avoid misinterpretations, we reformulated the text in Section 2.4 (p. 8, l. 17-21) in a revised version of the manuscript: "[…]. The model returns a probabilistic representation of the target value. If the model was trained on all available data, and is applied within the domain of these data, the predictions will be unbiased and neither over- nor underconfident. If instead a model using deterministic functions is trained and applied in the same manner, the resulting single-valued predictions may also be unbiased, but due to their single-value nature will surely be overconfident.".

**Minor Comments:**

**Comment 5**: Neither information theory nor curse of dimensionality need to be capitalized throughout.
**Response 5**: Thanks. We adapted the writing style throughout the text in a revised version of the manuscript.

**Comment 6**: p.1 l.23: In the abstract it is unclear what "relative magnitude of discharge in a 65-hour time window" means. "Relative" to what?
**Response 6**: We mean "relative to all values in the time window". We suggest rephrasing the sentence "[…]: discharge from two distinct time steps, the relative magnitude of discharge compared to all discharge values in a surrounding 65-hour time window, and event predictions

from the previous time step. We included our reformulated sentence in a revised version of the manuscript (Abstract, p. 1, l. 22 and Section 5, p. 19, l. 25).

**Comment 7**: p.2 l.3: The quote from Chow 1988 appears to be missing a word, "... physiographic and climatic [word missing] that govern..."
**Response 7**: Thank you. The correct quote is indeed "[...] physiographic and climatic characteristics that govern [...]". We included the correct quote in a revised version of the manuscript (Section 1, p. 2, l. 2).

**Comment 8**: p.2 l.5-7: Aren't i) and iv) basically the same?
**Response 8**: We rethought the arguments made here and suggest rephrasing the sentence: "Discharge time series are a fundamental component of hydrological learning and prediction since they i) are relatively easy-to-obtain, available in high quality and from widespread and long-existing observation networks. ii) carry robust and integral information about the catchment state, and iii) are an important target quantity for hydrological prediction and decision-making.". We included the reformulated sentence in a revised version of the manuscript (Section 1, p. 2, l. 5-7).

**Comment 9**: p.2 l.20-29: The discussion of the history of event detection doesn't provide a particularly historic view. Event-detection (and baseflow separation) have a history that goes back a lot further than 2006.
**Response 9**: Correct. In our discussion we focused on relatively recent techniques suitable for automated event detection. In order to put the paper in the proper historical context, we suggest adding in a revised version of the manuscript a brief overview on older methods of event detection and baseflow separation. We added a brief overview of older methods in Section 1, p. 2, l. 21-27).

**Comment 10**: p.3 l.11-15: Claims about the bias and confidence from data driven methods need citation.
**Response 10**: We suggest rephrasing to clarify what we wanted to say: "Predictions based on probabilistic models that learn relations among data directly from the data, with few or no prior assumptions about the nature of these relations, are less bias-prone (because there are no prior assumptions potentially obstructing convergence towards observed mean behavior), and less likely to be overconfident compared to established models (because applying deterministic models is still standard hydrological practice, and they are overconfident in all but the very few cases of perfect models). This applies at least if there is sufficient data to learn from, appropriate binning choices were made and the application remains within the domain of the data that were used for learning.". We suggest that these claims are self-evident and do not require a citation. We included the reformulated sentence with some minor adjusts in a revised version of the manuscript (Section 1, p. 3, l. 15-21).

**Comment 11**: p.4 l.1: It is unclear what is meant by "1:1" mapping between target and predictor data means on page 4 line 1.

**Response 11**: It means that each target has exactly one corresponding predictor, i.e., one particular value of target is related to one particular value of predictor (in contrast to 1:n or n:m relationships). We included a more elaborate explanation in a revised version of the manuscript (Section 2.1, p. 4, l. 10-11).

**Comment 12**: p.4 l.21: What happens when the system is not stationary?
**Response 12**: If the system is non-stationary, i.e. system properties change with time, the inconsistency between the learning and the prediction situation will result in additional predictive uncertainty. The problems associated with predictions of non-stationary systems apply to all modeling approaches. If a stable trend can be identified, a possible countermeasure is to do learning and prediction on detrended data and then reimpose the trend in a post-processing step. We added a non-stationarity discussion in Section 2.2 (p. 5, l. 3-6) in the revised version of the manuscript.

**Comment 13**: Equation 2 has an errant dot before $\log_2$.
**Response 13**: Thank you. We corrected the errant dot in Eq. (2), Section 2.3.1.

**Comment 14**: Page 6 line 26: good -> well.
**Response 14**: Thank you. We corrected this miswriting in a revised version of the manuscript (Section 2.3.2, p. 7, l. 7).

**Comment 15**: Section 2.3.1: The phrase "over the same underlying set of events" is unclear.
**Response 15**: We agree that the meaning is not clear. We suggest rephrasing "It is also possible to compare two probability distributions p and q.". We included the reformulated sentence in a revised version of the manuscript (Section 2.3.1, p. 6, l. 1).

**Comment 16**: The last sentence of page 6 requires a citation as well as clarification of what is meant by "how hard the Curse of Dimensionality hits"
**Response 16**: We have explained the Curse of Dimensionality in detail and with citations in the same section (p. 6 l.12-23), so we think there is no need to further explain it at the end of page 6. However, to make clearer what we mean we suggest rephrasing the sentence to "[…] i.e., it is a measure of the impact of the Curse of Dimensionality.". We included the reformulated sentence in a revised version of the manuscript (Section 2.3.2, p. 7, l. 14-15), but not extra information.

**Comment 17**: There are multiple (consistent) definitions of cross entropy.
**Response 17**: Sorry, we do not understand this comment. In the text we provide several definitions and interpretations of cross entropy to provide the reader with a comprehensive perspective of the subject. No action was taken regarding this comment.

**Comment 18**: p.7 l.24-25: How do you determine whether "appropriate binning choice were made"?
**Response 18**: A befitting binning choice is indeed a subject of ongoing research, see e.g. Gong et al. (2014) or Pechlivanidis et al. (2016). In general terms, as mentioned in the manuscript

(page 4, lines 8-16) an appropriate binning choice yields bins which are neither too narrow (which leads to a overfitted model) nor too wide (which leads to overly smoothed histograms, which will introduce a significant amount of bias and also discards information about the high resolution details of the distribution). Same as response #4 and #10. We reformulated the text in Section 2.4 (p. 8, l. 17-21) and create a reference for "appropriate binning choice" in (Section 1, p. 3, l. 20-21).

**Comment 19**: p.8 l.2: the area of the catchment should be in $km^2$.
**Response 19**: Sorry, we do not understand this comment. The catchment area is already expressed in km^2. No action was taken regarding this comment.

**Comment 20**: p.8 l.7: The arguments of the KL divergence need to be explained.
**Response 20**: Sorry, we do not understand this comment. We could not find arguments regarding KL divergence in the mentioned page. No action was taken regarding this comment.

**Comment 21**: Throughout section 3 discharge units should be $m^3 s^{-1}$
**Response 21**: Sorry, we do not understand this comment. Discharge values are already expressed in m^3 s^-1. No action was taken regarding this comment.

**Comment 22**: Section 3.1: How is snow taken into account in the discharge time series (since this changes the timing between precipitation and streamflow)?
**Response 22**: Please note that referee #1 had a similar question, so we mainly provide the same reply here as to referee #1, comment 5.
Effects of snow accumulation and melting on discharge events were not the central point of the study. Thus, they were not explicitly considered or classified by the model.
One of the strengths of the data-based approach we describe is that it potentially accepts any kind of additional predictors such as air temperature, nitrate concentrations, or snow. We agree with the referee that snow-related observations could be a potentially very useful predictor to identify snowmelt events. However, we suggest that doing so would add another facet of application to the manuscript, but would not add to the method description as such, which is the main goal of our paper. As mentioned in the responses #1 and #5 of referee #1. We added to the new version of the manuscript a brief discussion on the potential of using aggregated precipitation as a predictor and snowmelt events (Section 4.1.3, p. 17, l. 4-8).

**Comment 23**: Section 3.2.5: One sentence sections are rather odd.
**Response 23**: We agree that this looks odd. To avoid it we considered merging subsections of section 3.2. However, due to the nature of the different predictors (many discharge-based predictors and few others), in order to merge precipitation we'd have to merge them all, which would greatly reduce readability. We think that the one-sentence section is the lesser evil and suggest keeping the subsection structure in 3.2 as it is. No action was taken regarding this comment. Note that since the structure of the revised manuscript was rethought to settle the comparative investigation between ITM and CPM, the mentioned Section 3.2.5 is currently the Section 3.2.1.5 in the revised version.

**Comment 24**: In the application of the method to a new time series, what happens when you encounter conditions that did not previously occur and which are outside the range of your empirical PMF?

**Response 24**: Good point. If the conditions are outside of the range of the empirical PMF, the model will fail, i.e. it will provide no prediction. The same problem will also occur if the conditions are within the range of the empirical PMF, but have never been observed, at least not as seen through the filter of the chosen binning. In that case, the predictive distribution of the target (event Yes/No) will also be empty. If 'no answer' by the model is not acceptable, several methods exist to guarantee an answer, however at the cost of reduced precision:

- Coarse graining: The PMF can be rebuild with fewer, wider bins and an extension of the range (e.g. by merging neighboring bins) until the model provides an answer to the predictive setting. This way the PMF is still purely based on the observed data, but the resulting predictive distributions will be more spread-out as they integrate more observations from a larger range of observed situations. Similar methods to avoid empty bins by adjusting the binning have been proposed for example by Darbellay and Vajda (1999), Knuth (2013) and Pechlivanidis et al. (2016).
- Gap-filling: A widely applied alternative is to maintain the binning and fill the empty bins with non-zero values based on a deemed-to-be-reasonable assumption on the shape of the PMF if more data were available. Approaches comprise adding one counter to each zero-probability bin of the sample histogram, adding a small probability to the sample PDF, smoothing methods such as Kernel-density smoothing (Blower et al. , 2002; Simonoff, 1996), or Bayesian approaches based on the Dirichlet and Multinomial distribution, or a Maximum-Entropy Method recently suggested by Darscheid et al. (2018). All of these methods are applicable both for the extrapolation case mentioned by the referee and the interpolation case.

We included to the revised version of the manuscript a short discussion of this topic (Section 2.4, p. 8, l. 22-32).

**Comment 25**: The heading for section 4.2 is vague (non-descriptive)

**Response 25**: We chose this section header in relation to that of section 4.1 (model performance for the full data set) to emphasize the difference between the two. In order to maintain this emphasis, we prefer to keep the header as it is. Thus, no action was taken regarding this comment.

**Comment 26**: p.12 l.19: "Computationally expensive" - what does that mean in this context? 2 minutes on a laptop or a week on a 30 thousand core computing cluster?

**Response 26**: In our work it means one standard PC running for around 300 hours to analyze a 5- dimensional model (18 different sample sizes, 500 repetitions of each sample size). As it is well known that resampling-based analysis techniques tend to be computationally expensive, we suggest keeping the statement in the text as general as it is. Thus, no action was taken regarding this comment.

**Comment 27**: Conclusions: The first part of the conclusions is just a summary of the paper. I think this can be shortened.

**Response 27**: We agree that the summary can be shortened and suggest to do so in a revised version of the manuscript. However, we think it is important to give the reader a short wrap-up of what was done before the conclusions are drawn. We will not abandon the summary altogether. Also, in order to better reflect the content of the section, we suggest changing the section header from 'Conclusions' to 'Summary and conclusions'. We revised the manuscript conclusion (Section 5), renaming the header to 'Summary and conclusions', removing few excerpts of the summary, and including the cross-validation and comparative investigation procedure and results.

**16 | $Q(t), Q_{\text{RMC}}, Q(t+2)$**
**27 | - |**
| $P$-based group | - | $Q(t), P$
**23 | $Q(t), P, Q_{\text{RMC}}$**
**28 | - |**
| Model-based group
with $Q$-based predictors | - | - | - | $Q(t), Q_{\text{RMC}}, Q(t+2), e_{\text{P}_{\#27}}(t-1)$
**29 |**
| Model-based group
with $P$-based predictors | - | - | - | $Q(t), P, Q_{\text{RMC}}, e_{\text{P}_{\#28}}(t-1)$
**30 |**

**Table 7: Application I – Curse of Ddimensionality and data size validation for models in Table 6.**

| # | Predictive model | H(X) [bit] | H(X)/H(X)* | Sample size where $D_{KL}/H(X) \leq 5$ % and % of the full data set** | Sample size [a] | Number of bins |
|---|---|---|---|---|---|---|
| 0 | $e$ | 0.516 | 100 % | $\geq 4398$ (5.6 %) | 0.5 | 2 |

| # | Predictive model | H(X\|Y) [bit] | H(X\|Y)/H(X)* | Sample size where $D_{KL}/H(X\|Y) \leq 5$ % and % of the full data set** | Sample size [a] | Number of bins |
|---|---|---|---|---|---|---|
| 3 | $e \mid Q(t)$ | 0.260 | 50.4 % | $\geq 9952$ (12.6 %) | 1.1 | 68 |
| 16 | $e \mid Q(t), Q_{RMC}$ | 0.182 | 35.3 % | $\geq 29\,460$ (37.3 %) | 3.4 | 748 |
| 23 | $e \mid Q(t), P$ | 0.248 | 48.2 % | $\geq 18\,880$ (23.9 %) | 2.2 | 2108 |
| 27 | $e \mid Q(t), Q_{RMC}, Q(t+2)$ | 0.144 | 28.0 % | $\geq 60\,178$ (76.3 %) | 6.9 | 25\,432 |
| 28 | $e \mid Q(t), P, Q_{RMC}$ | 0.167 | 32.5 % | $\geq 50\,377$ (63.8 %) | 5.8 | 23\,188 |
| 29 | $e \mid Q(t), Q_{RMC}, Q(t+2), e_{p_{\#27}}(t-1)$ | 0.114 | 22.2 % | $\geq 69\,102$ (87.6 %) | 7.9 | 279\,752 |
| 30 | $e \mid Q(t), P, Q_{RMC}, e_{p_{\#28}}(t-1)$ | 0.142 | 27.6 % | $\geq 62\,667$ (79.4 %) | 7.2 | 255\,068 |

* H(X) = H(e) = 0.516 bits
** size of the full data set: 78 912 data points (9 years)

**Table 8: Cross-validation data set – Characteristics of the user event classification set.**

| Data set | Time steps classified as positive events $(P)$ | Time steps classified as non-events $(N)$ | Percentage of events $(P/T)$ | Percentage of non-events $(N/T)$ | Total $(T)$ |
|---|---|---|---|---|---|
| Training | 8150 | 60 952 | 11.8 % | 88.2 % | 69 102 |
| Testing | 942 | 8868 | 9.6 % | 90.4 % | 9810 |
| Sum | 9092 | 69 820 | 11.5 % | 88.5 % | 78 912 |

**Table 9: Application II – ITM and CPM performance.**

| Event detection method | True Positive $(P_T)$ | $R_{TP}$ $(P_T/P*)$ | False Positive $(P_F)$ | $R_{FP}$ $(P_F/N*)$ | Accuracy % $((P_T + N_T**)/(P*+N*))$ | Eq. (7) distance*** |
|---|---|---|---|---|---|---|
| ITM | 911 | 96.7 % | 1038 | 11.7 % | 89.1 % | 0.12 |
| CPM | 796 | 84.5 % | 877 | 9.9 % | 89.6 % | 0.19 |

\* $P = 942, N = 8868$ (Table 8)
\*\* $N_T = N - P_F$
\*\* Distance to the perfect model of the ROC curve

[Figure]

**Figure 1: Main steps of the ITMdata-driven model based on Information Theory.**

[Figure]

**Figure 2: Investigating the effect of sample size through Cross Entropy and Kullback-Leibler Divergence.**

[Figure]

**Figure 3: Input data – Discharge , precipitation , user-based event classification . Overview of the time series on the left and detailed example on the right.**

[Figure]

Figure 4: Window size definitions for window types. a) $Q_{RMC}$, b) $Q_{RML}$, c) $Q_{RMR}$ window definitions, and d) window size analysis.

[Figure]

**Figure 5: Cross Entropy for models in Table 6 as a function of sample size.**

[Figure]

Figure 6: Curse of Dimensionality for models in Table 6 measured by the ratio of $D_{KL}$ and $H(X|Y)$.

[Figure]

**Figure 76: Application I – Probabilistic prediction of 4-predictor model #29 (Table 5) to a subset of the training data.**

[Figure]

Figure 7: Application II – Binary prediction of ITM and CPM to a subset of the testing set.

[Figure]

**Figure A1: Dispersion analysis of the Cross Entropy. The effect of the number of repetitions in the target model (#0 in Table 7).**

---

## Author Response (AR2)

Dear Mrs. Schaefli,

The current document consolidates the referee comments and the author responses. After this section of comments and responses, please find the manuscript pdf with the tracked revision.

Best regards,
Stephanie Thiesen, Uwe Ehret and Paul Darscheid

**Referee #1 | Dr. Yiwen Mei**

Minor revisions

**Comment 1:** Section 3.2.1: In my opinion, there is no need to have more higher-level hierarchy in the section. I suggest to merge all those sub-sections under section 3.2.1. Same applied for section 3.2.3.

**Response 1**: The subsections under section 3.2.1 are meant to clearly separate and highlight the potential predictors in the text, and the authors prefer to keep as it is. The same idea of separation is applicable to subsections under section 3.2.3, which are individually mentioned in section 4.1.1 (p.14, l.1 and 12). Thus, considering that there is no gain of clarity with the suggested alteration, the authors prefer to keep the sections the way they are presented.

**Comment 2:** Title of Section 3.2 and 3.3: I would suggest better titles like for example "Information theory method" and "Characteristic point method", respectively, for section 3.2 and 3.3.

**Response 2**: We would like to keep Application I and II in the title, since these two applications have different purposes. The first one is probabilistic, while the second one is deterministic. However, since Application II involves the comparison between ITM and CPM (not just CPM), we suggest the following titles, keeping the acronym for sake of brevity:
3.2 Application I - ITM
3.3 Application II - ITM and CPM comparison

**Comment 3:** Section 4: I would suggest to keep only 2 hierarchy here. The current section 4.1 seems redundant.

**Response 3**: The authors believe that section 4.1 helps to keep the results from Application I (4.1, probabilistic ITM) and II (4.2, deterministic ITM and CPM) clearly separated. Additionally, subsections "4.1.1 Model performance for the full data set", "4.1.2 Model performance for samples" and "4.1.3 Model application" discuss different steps of Application I. Subsection 4.1.1 presents and compares the performance of the predictive models as a whole, subsection 4.1.2 presents the performance of the model for different sizes of data set, aiming at the model validation (validation of the size of the data set, and indirectly the binning choice, according to the predictive model selected in 4.1.1), and, finally, subsection 4.1.3 uses the results from the previous subsections to select the best model and proceed with its application. Thus, to keep these steps clear, we prefer to keep as it is.

**Comment 4:** Table 6: "Model-based group with $Q$-based predictors" is not accurate. Perhaps call it "Model-based group with sorely $Q$-based predictors".

**Response 4**: The meaning of the groups is previously introduced in subsection 3.2.3.1, however we agree that the current Table 6 can reinforce the information. We suggest the following footnote in Table 6 "* Models which apply exclusively discharge-based predictor(s) ** Models which apply discharge- and precipitation-based predictor(s)".

**Comment 5:** P15 line 4: insert "." After "repetitions".

**Response 5**: Thank you. We will correct this in a revised version of the manuscript.

**Comment 6:** Figure 7: I am not sure what criteria the "expert" used in manually identification of events. If I were the expert, I would group the last four events as two bi-model events. I feel like both ITM and CPM are doing better than the "expert" in these cases.

**Response 6**: As mentioned in the manuscript (p. 19, l. 8-12), the identification of events made by an expert is hard to reproduce since it dependents on acuity and knowledge of the event identifier. It can also vary depending on the expert interest and, since it is cumbersome for long time series, it is subject to handling errors. In this sense, the idea of the ITM method is to reproduce the main patterns of the expert identification (on which the model was trained) to mitigate these issues. Thus, based on the overall training data set, the model identified the patterns of the last events as two bi-model events, as mentioned by the referee.

**Comment 7:** P18 line 34: I would like to point out that the reason that CPM require precipitation data is because we need to study runoff coefficient and time lag in our original application. Without precipitation data, one can still use CPM to produce flow event period. This was implemented by Shen et al. (2017, Comprehensive Database of Flood Events in the Contiguous United States from 2002 to 2013) which developed a three-level flood event database. To produce the level 1 events, only the streamflow records were used and only the timing (begin, peak(s), end) of flood events were provided; level 3 events have all the event properties (e.g. runoff coefficient, event time lag, etc.,) that require the use of precipitation to compute. As the developer of the CPM program, I would say that the toughest aspect of implementing CPM could be the proper selections of model parameters. Also, I would like to take this chance to thank the authors for their idea of using a "synthetic true event database" to calibrate the model parameters. Though in my philosophy visual inspection by human is never a synthetic true, but it is a very clever attempt.

**Response 7**: For our analysis, we used the publicly available script for CPM, which has filters based on precipitation data implemented after the flow event identification. In the case analyzed in the paper, for example, the application of rainfall event-related filters considerably increased the quality of results: reducing the false positive rate from 12.2% (without rainfall event-related filters, but filtering out the small peak flow events) to 9.9% (with rainfall event-related filters), and the distance to the perfect model (Eq. 7) from 0.20 to 0.18 in the testing phase. Thus, since the code available uses the mentioned inputs (precipitation, catchment area and discharge) and the precipitation data assist in better results, the authors applied the method as it is. However, considering that the application is also possible without this data, we suggest rephrasing the sentence to:

"An interesting conclusion is that ITM was able to overcome CPM requiring only discharge data and a training data set of classified events (also based on the discharge set), while CPM demanded precipitation, catchment area and discharge as inputs. It is important to note that CPM can be modified to be used without precipitation data, however in our case it resulted in a considerably higher false positive rate, since the rainfall event-related filters cannot be applied.".

**Extras:** In addition to the changes suggested by the referee, the authors performed minor corrections concerning spacing, comma and language throughout the text. Complementarily, Table 9 and Figure 7 were updated, according to a script adjustment in the application of ITM. It should be noted that these adjustments did not qualitatively modify the results and conclusions presented.

We thank again the referees for cautiously reviewing our article and providing their feedbacks. Stephanie Thiesen, Uwe Ehret and Paul Darscheid

[revised manuscript text omitted]

* Models which apply exclusively discharge-based predictor(s)

** Models which apply discharge- and precipitation-based predictor(s)

**Table 7: Application I – Curse of dimensionality and data size validation for models in Table 6.**

| # | Predictive model | $H(X)$ [bit] | $H(X)/H(X)^*$ | Sample size where $D_{KL}/H(X) \leq 5\%$ and % of the full data set** | Sample size [a] | Number of bins |
|---|---|---|---|---|---|---|
| 0 | $e$ | 0.516 | 100 % | $\geq 4398$ (5.6 %) | 0.5 | 2 |

| # | Predictive model | $H(X\|Y)$ [bit] | $H(X\|Y)/H(X)^*$ | Sample size where $D_{KL}/H(X\|Y) \leq 5\%$ and % of the full data set** | Sample size [a] | Number of bins |
|---|---|---|---|---|---|---|
| 3 | $e \mid Q(t)$ | 0.260 | 50.4 % | $\geq 9952$ (12.6 %) | 1.1 | 68 |
| 16 | $e \mid Q(t), Q_{\mathrm{RMC}}$ | 0.182 | 35.3 % | $\geq 29\,460$ (37.3 %) | 3.4 | 748 |
| 23 | $e \mid Q(t), P$ | 0.248 | 48.2 % | $\geq 18\,880$ (23.9 %) | 2.2 | 2108 |
| 27 | $e \mid Q(t), Q_{\mathrm{RMC}}, Q(t+2)$ | 0.144 | 28.0 % | $\geq 60\,178$ (76.3 %) | 6.9 | 25\,432 |
| 28 | $e \mid Q(t), P, Q_{\mathrm{RMC}}$ | 0.167 | 32.5 % | $\geq 50\,377$ (63.8 %) | 5.8 | 23\,188 |
| 29 | $e \mid Q(t), Q_{\mathrm{RMC}}, Q(t+2), e_{\mathrm{p}_{\#27}}(t-1)$ | 0.114 | 22.2 % | $\geq 69\,102$ (87.6 %) | 7.9 | 279\,752 |
| 30 | $e \mid Q(t), P, Q_{\mathrm{RMC}}, e_{\mathrm{p}_{\#28}}(t-1)$ | 0.142 | 27.6 % | $\geq 62\,667$ (79.4 %) | 7.2 | 255\,068 |

* $H(X) = H(e) = 0.516$ bits
** size of the full data set: 78 912 data points (9 years)

**Table 8: Cross-validation data set – Characteristics of the user event classification set.**

| Data set | Time steps classified as positive events (*P*) | Time steps classified as non-events (*N*) | Percentage of events (*P/T*) | Percentage of non-events (*N/T*) | Total (*T*) |
|---|---|---|---|---|---|
| Training | 8150 | 60 952 | 11.8 % | 88.2 % | 69 102 |
| Testing | 942 | 8868 | 9.6 % | 90.4 % | 9810 |
| Sum | 9092 | 69 820 | 11.5 % | 88.5 % | 78 912 |

**Table 9: Application II – ITM and CPM performance.**

| Event detection method | True Positive $(P_T)$ | $R_{TP}$ $(P_T/P^*)$ | False Positive $(P_F)$ | $R_{FP}$ $(P_F/N^*)$ | Accuracy % $((P_T + N_T^{**})/(P^*+N^*))$ | Eq. (7) distance*** |
|---|---|---|---|---|---|---|
| ITM | 918 | 97.5 % | 1138 | 12.6 % | 88.4 % | 0.13 |
| CPM | 796 | 84.5 % | 877 | 9.9 % | 89.6 % | 0.18 |

\* $P = 942$, $N = 8868$ (Table 8)

\*\* $N_T = N - P_F$

\*\* Distance to the perfect model of the ROC curve

[Figure]

**Figure 1: Main steps of the ITM.**

[Figure]

**Figure 2: Investigating the effect of sample size through Cross Entropy and Kullback-Leibler Divergence.**

[Figure]

**Figure 3: Input data – Discharge, precipitation, user-based event classification. Overview of the time series on the left and detailed example on the right.**

[Figure]

**Figure 4: Window size definitions for window types. a) $Q_{RMC}$, b) $Q_{RML}$, c) $Q_{RMR}$ window definitions, and d) window size analysis.**

[Figure]

**Figure 5: Cross Entropy for models in Table 6 as a function of sample size.**

[Figure]

**Figure 6: Application I – Probabilistic prediction of 4-predictor model #29 (Table 5) to a subset of the training data.**

[Figure]

**Figure 7: Application II – Binary prediction of ITM and CPM to a subset of the testing set.**

[Figure]

**Figure A1: Dispersion analysis of the Cross Entropy. The effect of the number of repetitions in the target model (#0 in Table 7).**